# Minimizing False-Positive Attributions in Explanations of Non-Linear Models

**Anders Gjølbye**[1]    **Stefan Haufe**[2,3,4]    **Lars Kai Hansen**[1]
[1]Technical University of Denmark    [2]Technische Universität Berlin
[3]Physikalisch-Technische Bundesanstalt, Berlin    [4]Charité – Universitätsmedizin Berlin
agjma@dtu.dk    haufe@tu-berlin.de    lkai@dtu.dk

## Abstract

Suppressor variables can influence model predictions without being dependent on the target outcome, and they pose a significant challenge for Explainable AI (XAI) methods. These variables may cause false-positive feature attributions, undermining the utility of explanations. Although effective remedies exist for linear models, their extension to non-linear models and instance-based explanations has remained limited. We introduce PatternLocal, a novel XAI technique that addresses this gap. PatternLocal begins with a locally linear surrogate, e.g., LIME, KernelSHAP, or gradient-based methods, and transforms the resulting discriminative model weights into a generative representation, thereby suppressing the influence of suppressor variables while preserving local fidelity. In extensive hyperparameter optimization on the XAI-TRIS benchmark, PatternLocal consistently outperformed other XAI methods and reduced false-positive attributions when explaining non-linear tasks, thereby enabling more reliable and actionable insights. We further evaluate Pattern-Local on an EEG motor imagery dataset, demonstrating physiologically plausible explanations.

## 1   Introduction

In recent years, the growing demand for transparent, accountable, and ethical AI systems across various industries and research communities has driven substantial advancements in explainable artificial intelligence (XAI). The need to understand machine learning models' decision-making processes is well-established, particularly in high-stakes domains such as healthcare, finance, and criminal justice. While researchers have made promising progress with so-called self-explainable models such as prototypical networks, these approaches often fail to achieve the same downstream performance as conventional models and typically require complete retraining [22]. Consequently, researchers focus significant attention on *post-hoc* XAI methods that apply to existing models without retraining. Post-hoc XAI methods encompass a diverse array of approaches, including architecture-specific techniques such as Layer-wise Relevance Propagation (LRP) [1], concept-based explanations like TCAV [14], and model-agnostic frameworks such as SHAP [17] and LIME [21]. These methods promise to support model validation, data quality checks, and actionable intervention suggestions [23]. To provide evidence, researchers evaluate XAI methods against criteria such as faithfulness to the model, robustness to perturbations, and human interpretability [32]. However, none of these criteria guarantee that the features highlighted as important are discriminative for the target variable, an assumption behind many uses of XAI [13].

Several studies [34, 35] have shown that widely used XAI methods often assign importance to *suppressor variables* [9, 33], features that affect model predictions without having a direct statistical dependency with the target. Suppressors may capture correlations in noise or other auxiliary signals that models exploit to improve predictions without offering causal or correlational insights. For instance, a model might predict epilepsy from seizure activity in a specific brain region, but to

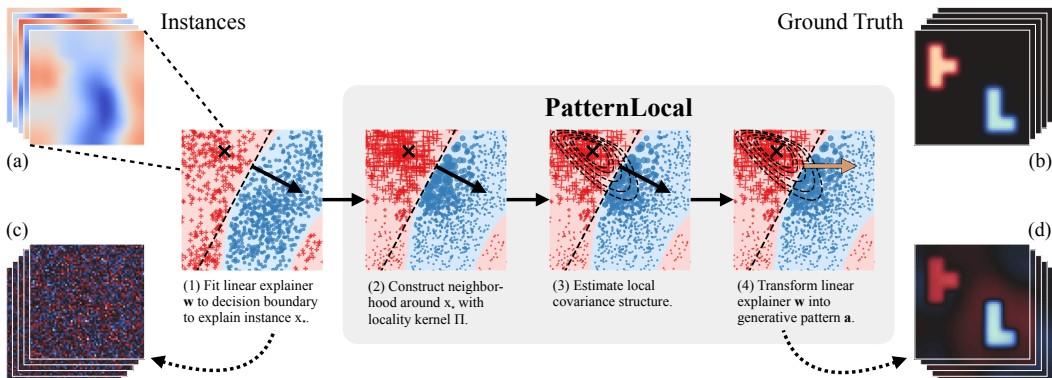

Figure 1: Conceptual step-by-step overview of how the PatternLocal method generates local explanations. A classifier is trained on the dataset (a), with both the dataset and the model's decision boundary visualized in a 2D projection. Many XAI methods produce importance maps (c) through local linearization (e.g., LIME, KernelSHAP). PatternLocal enhances this process by transforming the local discriminative surrogate into a generative explanation (d), thereby reducing the influence of irrelevant or misleading features. The resulting importance maps (c–d) can be directly compared with the ground-truth attributions (b).

optimally do so, it also relies on a noise probe from an unaffected region; common XAI methods may then highlight both regions, mistakenly assigning significance to the irrelevant area.

To support interpretations that rely on statistical relevance, a feature should depend on the target variable [34, 13]. The *Pattern* method [12] formalizes this idea by separating prediction from explanation: a discriminative model makes predictions, and a corresponding generative model provides explanations. Extensions such as PatternNet and PatternAttribution generalize this idea to deep neural networks [15] but have shown weaknesses on non-linear suppressor benchmarks like XAI-TRIS [8]. Unlike human-centered evaluations, benchmarks like XAI-TRIS provide controlled environments with objective ground-truth explanations to evaluate the robustness of XAI methods against suppressors and other uninformative variables. Also, model-agnostic methods such as SHAP, LIME, and LRP largely ignore training data structure, making them prone to suppressor bias [34, 35, 8].

In this work, we introduce *PatternLocal*, a data-driven, model-agnostic XAI method that extends the Pattern approach to non-linear cases. PatternLocal converts locally discriminative explanations, produced by methods such as LIME or KernelSHAP, into generative representations that more accurately reflect statistical relevance; see Figure 1. This transformation helps reduce the impact of suppressor and other non-informative variables while maintaining local fidelity to the model's decision behavior. PatternLocal assumes access to representative training data and a meaningful input simplification (e.g., superpixels or feature masks) shared across samples. We evaluate PatternLocal on the XAI-TRIS benchmark, artificial lesion MRI benchmark, and an EEG Motor imagery dataset, and compare it with a range of established XAI methods. The results show that PatternLocal consistently provides more reliable and interpretable explanations without changing the underlying model.

## 2 Preliminary

### 2.1 Notation

Let $\mathcal{X} \subset \mathbb{R}^D$ and $\mathcal{Y} \subset \mathbb{R}$ denote the input and output spaces, respectively. Consider a model $f : \mathcal{X} \to \mathcal{Y}$. For an instance $\mathbf{x}_\star \in \mathcal{X}$ we wish to explain the prediction $f(\mathbf{x}_\star)$. Because highly complex models such as deep networks are notoriously hard to interpret, they are approximated *locally* by a simpler, intrinsically interpretable model $g$. We therefore restrict attention to local explanations for the specific instance $\mathbf{x}_\star$ rather than to the behavior of $f$ on $\mathcal{X}$ as a whole.

Many XAI methods first replace $\mathbf{x}_\star$ with a lower-dimensional or more semantically meaningful representation $\mathbf{x}'_\star = h_{\mathbf{x}_\star}(\mathbf{x}_\star) \in \mathbb{R}^{D'}$. In image analysis, for example, $\mathbf{x}'_\star$ may encode superpixels; in other settings, it may correspond to a low-rank projection or simply a subset of raw features.

Such simplification often makes for easier interpretation and, when $D' \ll D$, also reduces the computational cost. See Appendix A for a summary of the notation.

**Remark.** In general, models that predict $y$ from $\mathbf{x}$ are termed *backward* or *discriminative*, whereas models that reconstruct $\mathbf{x}$ (or its sources) from latent factors are called *forward* or *generative*. We adopt this convention to avoid terminological clashes.

## 2.2 Suppressor Variables

Various studies [e.g., 9, 12] demonstrated that linear models may need to assign significant non-zero weight to correlated noise variables (suppressors). Given $\mathbf{x} \in \mathbb{R}^D$, assume each instance is generated from a linear model of the form

$$\mathbf{x} = \mathbf{A}\,\mathbf{m} + \boldsymbol{\varepsilon}, \qquad \mathbf{m} = \begin{bmatrix} m_1, \ldots, m_K \end{bmatrix}^\top \in \mathbb{R}^K, \tag{1}$$

where $K$ is the number of latent factors, $\mathbf{A} = [\mathbf{a}_1, \ldots, \mathbf{a}_K] \in \mathbb{R}^{D \times K}$ contains the *activation patterns* $\mathbf{a}_k$, and $\boldsymbol{\varepsilon}$ denotes additive noise. A linear backward model attempts to recover the latent vector $\mathbf{m}$ via

$$\hat{\mathbf{m}} = \mathbf{W}^\top \mathbf{x}, \qquad \mathbf{W} \in \mathbb{R}^{D \times K}. \tag{2}$$

Consider $D = 2$ and $K = 1$ with $x_1 = m + d$ and $x_2 = d$, where the scalar $m$ is the signal of interest and $d$ is a *suppressor*. The signal is exactly recovered by $\hat{m} = x_1 - x_2 = \mathbf{w}^\top \mathbf{x}$ with $\mathbf{w} = [1, -1]^\top$. Interpreting $\mathbf{w}$ directly as feature importance, however, suggests $x_1$ and $x_2$ contribute equally, even though the second channel carries only the suppressor, which is removed entirely in $\hat{m}$ by subtraction. This issue with suppressor variables extends to common explainable AI methods used to interpret machine learning models [35, 8].

**Remark.** While suppressor variables can improve prediction accuracy, they do not reflect genuine statistical dependence on the target. Such variables should be excluded from explanations, as they have no direct or indirect (causal, anti-causal, or confounded) link to the target. This differs from phenomena like the Clever Hans effect or shortcut learning, where models rely on spurious yet predictive features.

## 2.3 Forward vs. backward models

Instead of using the weights of the linear *backward* model, Haufe et al. [12] propose to use the corresponding *forward* model. They show that the weight matrix $\mathbf{W}$ has a unique forward model counterpart

$$\mathbf{A} = \Sigma_{\mathbf{X}}\,\mathbf{W}\,\Sigma_{\mathbf{M}}^{-1}, \tag{3}$$

where $\Sigma_{\mathbf{X}} = \mathrm{Cov}[\mathbf{x}]$ and $\Sigma_{\mathbf{M}} = \mathrm{Cov}[\hat{\mathbf{m}}]$ which recovers the activation pattern in Eq. (1). If the estimated factors are uncorrelated, $\Sigma_{\mathbf{M}}$ is diagonal and Eq. (3) reduces to $\mathbf{A} \propto \Sigma_{\mathbf{X}}\,\mathbf{W} = \mathrm{Cov}[\mathbf{x}, \hat{\mathbf{m}}]$. The resulting global feature importance map for the entire data set $\mathcal{D} \subset \mathcal{X}$ is

$$\mathbf{s}^{\text{Pattern}}(\mathcal{D}) = \Sigma_{\mathbf{X}}\,\mathbf{W}, \tag{4}$$

which mitigates suppressor variables for the linear model. For example, in unregularized linear discriminant analysis, where $\mathbf{W} = \Sigma_{\mathbf{X}}^{-1}(\boldsymbol{\mu}_+ - \boldsymbol{\mu}_-)$ with $\boldsymbol{\mu}_{+/-}$ denoting the class means, this approach completely removes the influence of suppressor variables in the activation pattern.

## 2.4 Local explainability methods

**LIME.** LIME works by constructing a local surrogate model $g$ of $f$ around $\mathbf{x}'_\star$; the surrogate's coefficients form the feature-importance explanation for $\mathbf{x}$ [21]. Its original formulation is $\xi = \operatorname{argmin}_{g \in G} \mathcal{L}(f, g, \Pi_{\mathbf{x}'_\star}) + \Omega(g)$, where $G$ is a class of interpretable functions, $\mathcal{L}$ is a loss evaluated on samples in the simplified space, $\Pi_{\mathbf{x}'_\star}$ is a local kernel, and $\Omega$ penalizes model complexity. In the common usage where $\mathcal{L}$ is squared loss and $g$ is linear, can be written as

$$\mathbf{s}^{\text{LIME}}(\mathbf{x}_\star) = \mathbf{w}^{\text{LIME}} = \operatorname*{argmin}_{\mathbf{v}} \mathbb{E}_{\mathbf{z}' \sim \mathbb{P}_{\mathcal{Z}}} \left[ \Pi_{\mathbf{x}'_\star}(\mathbf{z}') \big( f(h_{\mathbf{x}_\star}^{-1}(\mathbf{z}')) - \mathbf{v}^\top \mathbf{z}' \big)^2 \right] + \lambda\,R(\mathbf{v}), \tag{5}$$

where $R$ is a regularization weighted by $\lambda$, $\mathcal{Z}$ is the simplified input space, and $h_{\mathbf{x}_\star}^{-1} \colon \mathcal{Z} \to \mathcal{X}$ maps a simplified sample back to the original space. This notation is similar to the one by Tan et al. [31]. For

image explanations, the default choices are $\Pi_{\mathbf{x}'_\star}(\mathbf{z}') = \exp(-\|\mathbf{1} - \mathbf{z}'\|_0^2/\sigma^2)$, $R(\mathbf{v}) = \|\mathbf{v}\|_2^2$, and $h_{\mathbf{x}'_\star}^{-1}(\mathbf{z}') = \mathbf{z} = \mathbf{x}_\star \odot B(\mathbf{z}') + \mathbf{r} \odot (\mathbf{1} - B(\mathbf{z}'))$, with $\mathbf{z}' \sim \text{Uni}(\{0,1\}^{D'})$, $\odot$ denoting element-wise product, $r$ being a reference value, and where $B : \{0,1\}^d \to \{0,1\}^D$ copies each bit $z'_j$ to every pixel in superpixel. As mentioned, LIME can assign non-zero importance to suppressor variables [35].

**KernelSHAP & Gradient methods.** As shown by Lundberg and Lee [17], Tan et al. [31], KernelSHAP can be written in the form of Eq. (5) by setting $R(\mathbf{v}) = 0$ and $\Pi_{x'}(z') = (D' - 1)\big/\left[(D' \text{ choose } \|z'\|_0) \cdot \|z'\|_0 \cdot (D' - \|z'\|_0)\right]$. Tan et al. [31] further demonstrate that Eq. (5) unifies SmoothGrad [28] and the plain Gradient method [37].

# 3   PatternLocal

We now unify the Pattern insight that *forward* model parameters reveal statistically relevant features with the ability to probe highly non-linear models locally with methods like LIME. We call this *PatternLocal*. PatternLocal converts any linear surrogate explanation $\mathbf{w}$ produced around an instance $\mathbf{x}_\star$ into a data-driven pattern $\mathbf{a}$ that (1) suppresses false positives caused by suppressor variables, (2) remains faithful to the model because it builds on the very surrogate that approximates the decision boundary, (3) preserves local fidelity by weighting the surrounding data with a local kernel, and (4) remains practical in high-dimensional settings by operating in a simplified, low-dimensional space with regularized estimation.

**(1) Suppressor-variable mitigation.** PatternLocal mitigates suppressor effects by regressing the neighborhood of $\mathbf{x}_\star$ onto the local surrogate output $\tilde{y} = \mathbf{w}^\top \mathbf{x}$ under a Gaussian noise model. This reconstruction of $\mathbf{x}$ from $\tilde{y}$ effectively down-weights suppressor contributions.

**(2) Model faithfulness.** PatternLocal inherits $\mathbf{w}$ from Eq. (5), which approximates the model $f$ locally around $\mathbf{x}_\star$. By regressing the simplified input $\mathbf{x}$ on the surrogate response $\tilde{y}$ rather than the true label $y$, PatternLocal ensures that explanations describe the model's behavior, not the underlying task.

**(3) Local fidelity.** In non-linear models, the important features can vary across different regions of the input space. PatternLocal enforces locality by weighting all terms in its regression objective using the local kernel $\Pi_{\mathbf{x}'_\star}$. This ensures that the explanation focuses on the neighborhood around $\mathbf{x}_\star$ and does not reflect global structure.

**(4) Feasibility and dimensionality.** PatternLocal assumes sufficient local samples near $\mathbf{x}_\star$ and operates in the simplified space defined by $\mathbf{h}_{\mathbf{x}_\star}$, which aligns with Eq. (5) and reduces the dimensionality. To maintain stability in sparse or high-dimensional settings, the regression includes a regularization term $Q$.

## 3.1   Formal objective

Let $\mathbf{h}_{\mathbf{x}_\star} : \mathcal{X} \to \mathbb{R}^{D'}$ be the simplified input representation used in Eq. (5) and $\mathbf{w}$ be its solution. Also let $\tilde{y} = \mathbf{w}^\top \mathbf{h}_{\mathbf{x}_\star}(\mathbf{x})$ denote the local surrogate prediction. PatternLocal estimates a pattern $\mathbf{a}$ by regressing the local prediction onto the simplified input of the original data within the local neighborhood of $\mathbf{x}_\star$

$$\mathbf{s}^{\text{PatternLocal}}(\mathbf{x}_\star) = \mathbf{a} = \underset{\mathbf{u}}{\arg\min} \; \mathbb{E}_{\mathbf{x} \sim \mathbb{P}_\mathcal{X}} \left[ \Pi_{\mathbf{x}'_\star}(\mathbf{h}_{\mathbf{x}_\star}(\mathbf{x})) \left\| \mathbf{h}_{\mathbf{x}_\star}(\mathbf{x}) - \mathbf{u}\,\tilde{y} \right\|_2^2 \right] + \lambda Q(\mathbf{u}). \quad (6)$$

When $Q(\mathbf{u}) = \|\mathbf{u}\|_1$, the objective can be solved with a *kernel-weighted Lasso* regression. For $Q(\mathbf{u}) = \|\mathbf{u}\|_2^2$ the problem reduces to a *kernel-weighted ridge* regression with a closed-form solution given by

$$\mathbf{a}_{\ell_2} = \left( \mathbb{E}_{\Pi_{\mathbf{x}'_\star}}[\tilde{y}^2] + \lambda I \right)^{-1} \mathbb{E}_{\Pi_{\mathbf{x}'_\star}}\left[ \mathbf{h}_{\mathbf{x}_\star}(\mathbf{x})\,\tilde{y} \right] = \frac{\text{Cov}_{\Pi_{\mathbf{x}'_\star}}\left[ \mathbf{h}_{\mathbf{x}_\star}(\mathbf{x}), \tilde{y} \right]}{\text{Var}_{\Pi_{\mathbf{x}'_\star}}[\tilde{y}] + \lambda}, \quad (7)$$

where expectations, covariances, and variances are taken with respect to the normalized kernel $\Pi_{\mathbf{x}'_\star}(\mathbf{h}_{\mathbf{x}_\star}(\mathbf{x}))$ over $\mathcal{X}$. See Appendix B.1 for derivations. Eq. (7) therefore expresses $\mathbf{a}_{\ell_2}$ as the kernel-weighted covariance between the simplified features and the surrogate response, scaled by its regularized variance. Also see Appendix B.3 for more on sample complexity.

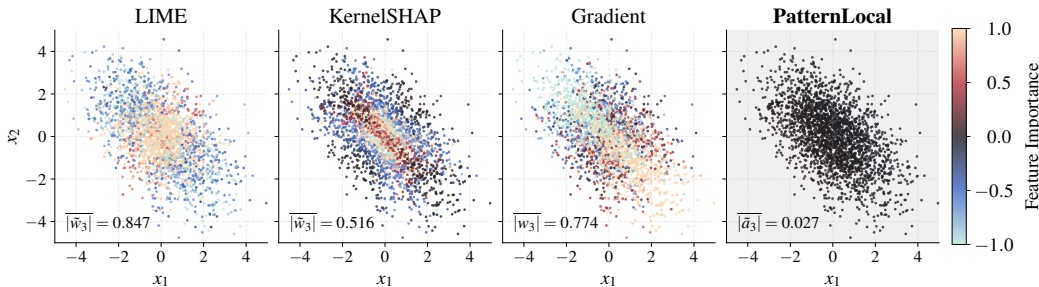

Figure 2: Feature–importance comparison on the XOR toy problem. We draw $2\,500$ i.i.d. samples from the generative process of Eq. (8), so the label depends solely on the interaction between $x_1$ and $x_2$, while $x_3$ is a *suppressor* that carries no marginal predictive signal. Each plot shows $(x_1, x_2)$ pairs, colored by the local normalized feature importance that four XAI methods assign to the suppressor variable $x_3$ when applied to the smooth classifier. Every plot reports the empirical mean magnitude of this attribution across all test points. Whereas LIME, KernelSHAP, and Gradient all attribute substantial importance to $x_3$, PatternLocal correctly drives the attribution of $x_3$ to (almost) zero.

## 3.2 Toy example

We present a small-scale toy example to better illustrate the intuition behind the PatternLocal method. Let $m_1, m_2 \sim \mathcal{N}(0, 1)$ and $d \sim \mathcal{N}(0, \sigma_d^2)$ be independent latent variables. We observe

$$\mathbf{x} = \begin{bmatrix} x_1 \\ x_2 \\ x_3 \end{bmatrix} = \begin{bmatrix} m_1 + d \\ m_2 - d \\ d \end{bmatrix}, \qquad y = \mathrm{sign}(m_1 m_2), \tag{8}$$

which induces a classical XOR-like dependency between $(x_1, x_2)$ and the label $y$. The variable $x_3$ acts as a suppressor. The optimal classifier for this non-linear problem is given by $\tilde{y} = \mathrm{sign}(\tilde{f}(\mathbf{x}))$, $\tilde{f}(\mathbf{x}) = (x_1 - x_3)(x_2 + x_3)$. A smooth approximation to this classifier is $f_\tau(\mathbf{x}) = \tanh((1/\tau)\,\tilde{f}(\mathbf{x}))$, $0 < \tau \ll 1$. Fixing an input $\mathbf{x}_\star$, the first-order (linear) approximation of the decision function at this point is $f_\tau(\mathbf{x}) \approx f_\tau(\mathbf{x}_\star) + \mathbf{w}^\top(\mathbf{x} - \mathbf{x}_\star)$ with

$$\mathbf{w} = \nabla f_\tau(\mathbf{x}_\star) = c_\star \begin{bmatrix} x_{2\star} + x_{3\star} \\ x_{1\star} - x_{3\star} \\ x_{1\star} - x_{2\star} - 2x_{3\star} \end{bmatrix}, \qquad c_\star = \frac{1}{\tau}\,\mathrm{sech}^2\left(\frac{1}{\tau}\tilde{f}(\mathbf{x}_\star)\right), \tag{9}$$

noting that $w_1 - w_2 + w_3 = 0$. Therefore, we generally have $w_3 \neq 0$, implying that local linear XAI methods (e.g., gradients, LIME, KernelSHAP) will incorrectly assign nonzero importance to the suppressor variable $x_3$.

Let $\Pi_{\mathbf{x}_\star}(\mathbf{x}) = \varphi(\|\mathbf{x} - \mathbf{x}_\star\|)$ be any isotropic local kernel. For $\lambda = 0$ and identity mapping $h_{\mathbf{x}_\star}(\mathbf{x}) = \mathbf{x}$, the PatternLocal explanation vector is defined as $\mathbf{a} = \mathrm{Cov}_{\Pi_{\mathbf{x}_\star}}[\mathbf{x}, \tilde{y}] / \mathrm{Var}_{\Pi_{\mathbf{x}_\star}}[\tilde{y}]$ and $\tilde{y} = \mathbf{w}^\top \mathbf{x}$.

$$\mathbf{a} = \frac{\mathrm{Cov}_{\Pi_{\mathbf{x}_\star}}[\mathbf{x}, \tilde{y}]}{\mathrm{Var}_{\Pi_{\mathbf{x}_\star}}[\tilde{y}]}, \qquad \tilde{y} = \mathbf{w}^\top \mathbf{x}.$$

Consider the third component $a_3 = \mathrm{Cov}_{\Pi_{\mathbf{x}_\star}}[x_3, \tilde{y}] / \mathrm{Var}_{\Pi_{\mathbf{x}_\star}}[\tilde{y}]$, we compute

$$\mathrm{Cov}_{\Pi_{\mathbf{x}_\star}}[x_3, x_1] = \sigma_d^2, \qquad \mathrm{Cov}_{\Pi_{\mathbf{x}_\star}}[x_3, x_2] = -\sigma_d^2, \qquad \mathrm{Cov}_{\Pi_{\mathbf{x}_\star}}[x_3, x_3] = \sigma_d^2.$$

Thus,

$$\mathrm{Cov}_{\Pi_{\mathbf{x}_\star}}[x_3, \tilde{y}] = w_1 \mathrm{Cov}_{\Pi_{\mathbf{x}_\star}}[x_3, x_1] + w_2 \mathrm{Cov}_{\Pi_{\mathbf{x}_\star}}[x_3, x_2] + w_3 \mathrm{Cov}_{\Pi_{\mathbf{x}_\star}}[x_3, x_3]$$
$$= w_1 \sigma_d^2 - w_2 \sigma_d^2 + w_3 \sigma_d^2 = \sigma_d^2(w_1 - w_2 + w_3) = 0,$$

and hence $a_3 = 0$. Similar calculation can be done for $a_1$ and $a_2$ which yields $\mathbf{a} \propto [w_1, w_2, 0]^\top$. See Appendix B.2 for derivations. We conclude that PatternLocal correctly assigns zero feature importance to the suppressor variable $x_3$, in contrast to standard local linear methods. This is also confirmed with simulation as seen in Figure 2.

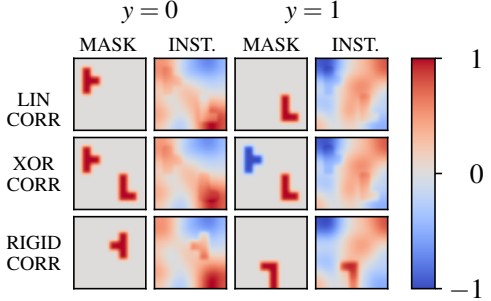 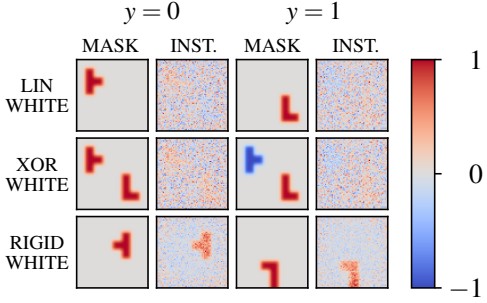

(a) Scenario CORR with $\alpha = 0.2$, where noise is correlated with the label, introducing a suppressor variable.

(b) Scenario WHITE with $\alpha = 0.2$, where noise is independent (white) thus no suppressor variable.

Figure 3: Examples of $64 \times 64$ instances from the XAI-TRIS Benchmark across six different scenario types. Each row represents a distinct structural pattern (LIN, XOR, RIGID), and each column pair shows a binary label class ($y = 0$ and $y = 1$) along with the corresponding instance and attribution mask. On the right (3a), correlated noise introduces a spurious suppressor effect, while on the left (3b), white noise does not.

## 4 Experiments

The experimental workflow begins with the construction of *synthetic* image datasets whose pixel-level ground-truth attribution maps are known by design, giving us complete control over class-conditional distributions.[1] We train each candidate classifier on a standard train/validation split and assess its performance on an unseen test set. For every test image, the selected XAI method produces a normalized importance map in the range $[-1, 1]$. These maps are then quantitatively compared with the ground-truth attributions using the evaluation metrics introduced below. All XAI hyperparameters are tuned on the validation split before the final assessment to ensure a fair comparison.

### 4.1 Datasets

We focus on the XAI-TRIS benchmark dataset [8] and the artificial lesion MRI dataset [19]. Both datasets specifically enable the assessment of XAI methods in the context of suppressor variables.

**XAI-TRIS Benchmark.** We follow the methodology outlined in Clark et al. [8] and provide complete details in Appendix C.1. The XAI-TRIS benchmark dataset consists of $64 \times 64$ images (with $N = 40,000$ samples) generated by combining a class-dependent *foreground* tetromino shape with *background* noise. The relative weighting between signal and noise is set by $\alpha$. We consider two noise scenarios:

- WHITE: uncorrelated Gaussian noise,
- CORR: spatially smoothed Gaussian noise (leading to *suppressor variables*).

We evaluate three scenarios:

- LIN: A 'T'-shaped tetromino near the top-left indicates $y = 0$; an 'L'-shaped tetromino near the bottom-right indicates $y = 1$.
- XOR: Both tetrominoes appear with differing signs depending on $y$.
- RIGID: Each tetromino is randomly translated and rotated by $90°$ increments.

The tetromino shape determines the set of important (ground-truth) pixels. For training and evaluation, each dataset is split into $\mathcal{D}_{\text{train}}$, $\mathcal{D}_{\text{val}}$, and $\mathcal{D}_{\text{test}}$ in a 90/5/5 ratio. Figure 3 shows examples of all scenarios. Spatially correlated noise in the CORR setting introduces *suppressor pixels*, background locations correlated with the foreground through the smoothing operator $G$. Although these pixels carry no direct class information, the model can exploit them to denoise the true signal [12, 34].

---

[1]Code is available at https://github.com/gjoelbye/PatternLocal.

**Artificial lesion MRI dataset.** Following Oliveira et al. [19], the Artificial Lesion MRI dataset contains grayscale medical downscaled images, with $D = 128 \times 128 = 16\,384$. It includes $N = 7\,500$ samples, evenly split into $\mathcal{D}_{\text{train}}$, $\mathcal{D}_{\text{val}}$, and $\mathcal{D}_{\text{test}}$. Each sample simulates a brain MRI slice with artificial lesions of predefined shapes and intensities, loosely resembling real-world white matter hyperintensities (WMH). Unlike XAI-TRIS, this dataset introduces challenges with more complex, organic lesion structures and anatomical background variations, making it a valuable benchmark for neuroimaging explainability. See Appendix C.2 for more details.

**EEG Motor Imagery dataset.** We use the PhysioNet EEG Motor Movement/Imagery dataset [24], containing recordings from 109 subjects performing motor tasks. EEG was acquired from 64 scalp electrodes at 160 Hz following the 10–10 system. We focus on runs R03, R07, and R11, where subjects repeatedly opened and closed their left or right fist. Labels T1 and T2 mark left- and right-fist movements, defining a binary classification problem ($y = 1$ for left, $y = 0$ for right). We use 19 standard 10–20 electrodes, resample to 100 Hz, apply a 0.5 Hz high-pass filter, and re-reference to the common average. A 3s window (1s before to 2s after onset) is extracted for each event, yielding 19 channels and 300 time points per trial. After removing invalid segments, the final dataset comprises $N = 4,927$ balanced trials (2,456 right, 2,471 left), split into $\mathcal{D}_{\text{train}}$, $\mathcal{D}_{\text{val}}$, and $\mathcal{D}_{\text{test}}$ in a 90/5/5 ratio.

## 4.2 Models

We use two types of classifiers for the synthetic datasets: a multi-layer perceptron (MLP) and a convolutional neural network (CNN). For the EEG Motor Imagery dataset, we use the ShallowFBCSPNet from Schirrmeister et al. [25]. Each classifier $f : \mathcal{X} \to \mathcal{Y}$ is trained on the training dataset $\mathcal{D}_{\text{train}}$. We train the models with early stopping based on validation performance on $\mathcal{D}_{\text{val}}$, and then evaluate them on the test set $\mathcal{D}_{\text{test}}$. We consider a classifier to have generalized to the classification problem if its test accuracy exceeds 90%. See Appendix D.1 for details on model architecture, training, and performance.

## 4.3 XAI methods

We evaluate nine methods on the models, including the PatternLocal approach and baselines. The XAI methods applied are: LIME [21], Integrated Gradients [30], Saliency [27], DeepLift [26], GradientShap [10], and GuidedBackProp [29]. Additionally, we incorporate two edge detection filters, Sobel and Laplace, which prior work shows outperform more advanced XAI techniques in suppressor variable scenarios [8]. Note that KernelSHAP is a special case of LIME. Since we perform extensive hyperparameter optimization, we may also select KernelSHAP during this process.

For the XAI-TRIS dataset, we benchmark LIME and PatternLocal with three input-simplification schemes for $\mathbf{h}_{\mathbf{x}_\star}(\cdot)$: the identity mapping, a superpixel representation, and a low-rank approximation. We evaluate the same three variants on the artificial-lesion MRI dataset. Superpixels are generated on a uniform grid for XAI-TRIS, whereas we use the slic algorithm for the MRI dataset.

## 4.4 Metrics

We evaluate explanation quality using two complementary metrics: the Earth Mover's Distance (EMD) and the Importance Mass Error (IME). See Appendix F.1 for a discussion of additional metrics such as faithfulness.

**Earth Mover's Distance (EMD).** The Earth Mover's Distance (EMD) measures the optimal cost to transform one distribution into another. For a continuous-valued importance map $s$ and ground truth $\mathcal{F}^+$, both normalized to have the same mass, we compute the EMD using the Euclidean distance as the ground metric. The optimal transport cost, $\text{OT}(|\mathbf{s}|, \mathcal{F}^+)$, is calculated following the algorithm proposed by Bonneel et al. [7] and implemented in the Python Optimal Transport library [11]. We define a normalized EMD performance score as

$$\text{EMD} = \text{OT}(|\mathbf{s}|, \mathcal{F}^+)/\delta_{max} \,, \tag{10}$$

where $\delta_{max}$ is the maximum Euclidean distance between any two pixels.

**Importance Mass Error (IME).**    As argued in Clark et al. [8] and Arras et al. [2], it is plausible that the given model only uses a subset of the important pixels for its prediction. Thus, false positives should be preferentially penalized while false negatives should be largely ignored. To this end, we provide an additional metric, Importance Mass Error (IME),

$$\text{IME} = 1 - \sum_{i=1}^{\text{card}(\mathcal{F}^+)} |s_i| / \sum_{i=1}^{\text{card}(\mathbf{s})} |s_i|. \tag{11}$$

## 4.5  Hyperparameter optimization

For both the synthetic *XAI-TRIS Benchmark* and the semi-synthetic *Artificial Lesion MRI* dataset, we conduct extensive hyperparameter optimization across all datasets, scenarios, SNR levels, and corresponding models to ensure a fair evaluation of all XAI methods. We employ Bayesian optimization using the Tree-structured Parzen Estimator (TPE) algorithm [4], implemented via the `hyperopt` package [5], which efficiently manages mixed continuous and categorical search spaces. The optimization objective minimizes the Earth Mover's Distance (EMD) between the absolute normalized explanation and the ground truth on the validation set. Additional implementation details are provided in Appendix D.2.

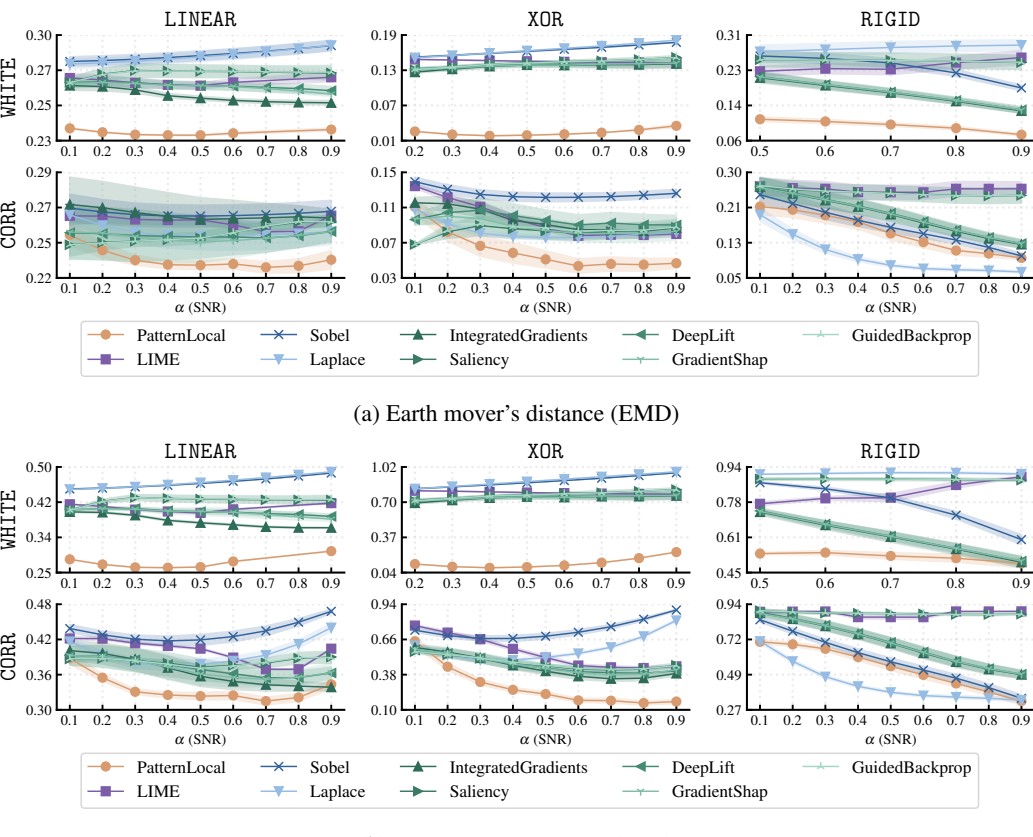

Figure 4: Quantitative evaluation of feature importance maps for the MLP model using $64 \times 64$ XAI-TRIS benchmark images generated by various methods, as a function of the SNR. The MLP model achieves at least $90\%$ accuracy, and the method's hyperparameters were optimized for EMD. The function $\mathbf{h}_{\mathbf{x}_\star}(\cdot)$ is the identity function, and $R(\cdot) = Q(\cdot) = \|\cdot\|_2^2$. The top panel (4a) shows results based on EMD, while the bottom panel (4b) presents performance in terms of IME. PatternLocal demonstrates significant improvements over most methods. Shaded regions indicate standard deviation.

# 5 Results

**XAI-TRIS Benchmark.** We evaluate the performance of the XAI methods on the XAI-TRIS benchmark both qualitatively and quantitatively. We conduct qualitative evaluation through visual inspection, which is standard in XAI research [e.g., 3, 17, 8], and illustrate it in Appendix E.1. The quantitative evaluation uses ground truth along with the EMD and IME metrics, as shown in Figure 4. The results presented here are for the MLP model, $\mathbf{h}_{\mathbf{x}_\star}(\cdot)$ as the identity function and $R(\cdot) = Q(\cdot) = \|\cdot\|_2^2$. Results for the CNN model, $\mathbf{h}_{\mathbf{x}_\star}(\cdot)$ as superpixels or low-rank approximation, and for $R(\cdot) = Q(\cdot) = \|\cdot\|_1$, are provided in Appendix E.1.

In Figure 4, we present the aggregated results of the methods across the test set for the EMD and IME metrics. Overall, the PatternLocal method substantially outperforms all other methods in the `XOR` and `RIGID` scenarios. However, we observe mixed results in the `RIGID CORR` scenario: PatternLocal performs substantially better than other XAI methods but is comparable to or worse than the filter methods Sobel and Laplace. These filter methods also performed well in Clark et al. [8] due to XAI-TRIS images having rigid bodies with sharp lines, making them especially easy for filter-based methods to detect. This advantage does not apply to images with more complex backgrounds, such as those in the artificial lesion MRI dataset.

Additional ablation studies on hyperparameters' impact and the suppressor variable's role are in Appendix G.

**Artificial lesion MRI dataset.** We evaluate the XAI methods using the same procedure on the artificial lesion MRI dataset. The results reported here correspond to the CNN model, with $\mathbf{h}_{\mathbf{x}_\star}(\cdot)$ defined over superpixels and $R(\cdot) = Q(\cdot) = \|\cdot\|_2^2$. Figure 5 illustrates the performance of *Pattern-Local* compared to LIME and the filter-based methods Sobel and Laplace. We provide additional qualitative and quantitative benchmark results in Appendix E.2.

**EEG Motor Imagery dataset.** Since the EEG motor imagery dataset does not have a ground-truth, we assess the explanations' physiological plausibility. This didactic example demonstrates how explanations can be meaningfully analyzed when they align with the underlying neurophysiology.

We used PatternLocal with the bandwidth set to the median pairwise distance, $\mathbf{h}_{\mathbf{x}_\star}(\cdot)$ as the identity function and no regularization. Applying PatternLocal to the motor-imagery EEG data produced explanations with clear temporal and spatial structure. The explanations concentrate on task onset and retain distinct oscillatory patterns in the alpha and gamma bands. When we apply spatio-spectral decomposition (SSD) [18] to single-trial explanations, the resulting components remain

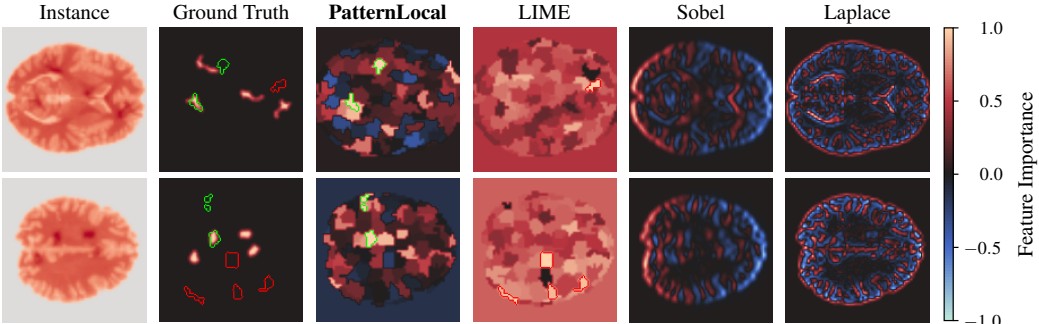

Figure 5: Examples from the artificial lesion MRI dataset, showing two MRI slices (rows). The first column displays the original MRI slices with artificially generated lesions; the second column (Ground Truth) shows the true lesion locations. The subsequent columns show feature importance maps from PatternLocal, LIME, Sobel, and Laplace. We use a CNN model, $\mathbf{h}_{\mathbf{x}_\star}(\cdot)$, with superpixels, and set $R(\cdot) = Q(\cdot) = \|\cdot\|_2^2$ for both PatternLocal and LIME. Note that the filter-based methods fail in these examples due to the lack of clear edges. Overlaid on the ground truth are superpixels with feature importance above 0.9 from PatternLocal (green) and LIME (red). PatternLocal explanations appear better aligned with the lesion locations. The overlap is not perfect, as the superpixel resolution limits it.

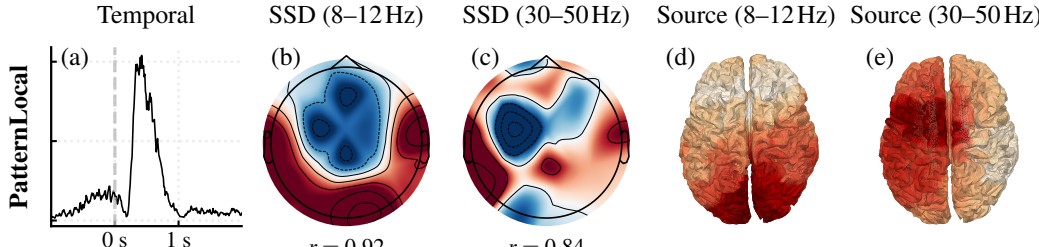

Figure 6: (a) Channel-averaged explanation signals of the PatternLocal explanation shows a marked increase around cue onset. (b-c) First SSD patterns derived from the explanation in the alpha (8-12 Hz) and gamma (30-50 Hz) bands have a good dipole fit (alpha r = 0.92, gamma r = 0.84). (d-e) eLORETA source localization of the SSD-reconstructed explanation highlights peri-rolandic cortex with stronger power in the left hemisphere, consistent with contralateral activation during right-hand imagery. SSD is applied directly to single-trial explanations before source reconstruction.

physiologically interpretable and localize with eLORETA[20] to the expected motor regions. For the right-hand example shown in Figure 6, the source reconstruction reveals increased left-hemisphere activity, reflecting contralateral motor engagement. Across the test set, dipole fits of SSD patterns indicate higher physiological plausibility for PatternLocal ($0.756 \pm 0.090$), with the raw instances showing similar results ($0.738 \pm 0.013$) and LIME performing worse ($0.604 \pm 0.013$). These findings demonstrate that PatternLocal yields explanations consistent with the physical forward model, enabling valid time-frequency and source analyses, unlike methods such as LIME, which do not permit meaningful decomposition of their outputs in this way.

## 6  Conclusion

This paper presents *PatternLocal*, a data-driven, model-agnostic XAI method that improves local explanations by reducing false-positive attributions caused by suppressor variables. By transforming standard local surrogate outputs into more reliable feature attributions, PatternLocal enhances interpretability without retraining the underlying model. Across three datasets, including XAI-TRIS, artificial lesion MRI, and EEG motor imagery, PatternLocal consistently outperforms existing methods, providing more precise and trustworthy explanations, particularly in challenging scenarios with misleading noise.

## Limitations

The primary limitation of PatternLocal is its reliance on access to representative training data in the neighborhood of each explanation. While this is feasible, it may be challenging in privacy-sensitive or data-scarce environments. Additionally, because PatternLocal fits a local generative model, it assumes a degree of consistency or alignment across samples in the input space. This assumption holds well in structured domains like medical imaging, where standardized views are common. However, it may break down in less constrained settings, such as natural images or user-generated content, where spatial alignment across instances cannot be guaranteed. PatternLocal reduces false-positive feature attribution; however, like other XAI methods, it can still misattribute importance, so saliency maps should be seen as suggestive rather than definitive.

## Acknowledgments

We thank Sebastian Weichwald, William Lehn-Schiøler, Teresa Dorszewski, and Lina Skerath for their contributions during the early stages of this work. We are also grateful to the anonymous reviewers for their valuable feedback. This work was supported by the Danish Data Science Academy, funded by the Novo Nordisk Foundation (NNF21SA0069429) and VILLUM FONDEN (40516), as well as the Pioneer Centre for AI (DNRF P1) and the Novo Nordisk Foundation ("Cognitive spaces – Next generation explainability", NNF22OC0076907).

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

# A    Notation

Table 1: Summary of notation.

| Symbol | Description |
| --- | --- |
| $\mathcal{X} \subset \mathbb{R}^D$ | Input space of dimension $D$. |
| $\mathcal{Y} \subset \mathbb{R}$ | Output space (scalar). |
| $\mathbf{x} \in \mathcal{X},\, y \in \mathcal{Y}$ | A data point and its label. |
| $\mathbf{x}_\star$ | Instance whose prediction $f(\mathbf{x}_\star)$ is explained. |
| $D,\, D'$ | $D$: input dimensionality; $D' \ll D$: simplified dimensionality. |
| $\mathbf{x}' = h(\mathbf{x})$ | Simplified/semantic representation of $\mathbf{x}$. |
| $\mathbf{h} : \mathcal{X} \to \mathbb{R}^{D'},\; \mathbf{h}^{-1} : \mathbb{R}^{D'} \to \mathcal{X}$ | Forward and inverse mapping between spaces. |
| $f : \mathcal{X} \to \mathcal{Y}$ | Original model being explained. |
| $g$ | Interpretable surrogate model fit locally around $\mathbf{x}_\star$. |
| $\Pi_{\mathbf{x}'_\star}(\cdot)$ | Local kernel weighting samples by proximity in simplified space. |
| $R(\cdot),\, Q(\cdot)$ | Regularization in LIME and PatternLocal. |
| $\lambda$ | Regularization strength. |
| $\alpha$ | Signal-to-noise mixing coefficient for synthetic data. |
| $\mathbf{z}',\, \mathcal{Z}$ | Simplified artificial sample and its space. |
| $\mathbf{s},\, \mathbf{s}^{\text{LIME}},\, \mathbf{s}^{\text{PatternLocal}}$ | Feature-importance maps. |
| $\mathbf{w}$ | Coefficient vector of the linear surrogate. |
| $\mathbf{a}$ | PatternLocal explanation vector via forward model regression. |
| $\mathbf{u},\, \mathbf{v}$ | Regression coefficients that minimize Eq. 5 or Eq. 6. |
| $\mathbf{W}$ | Backward weight matrix by the Pattern method. $\tilde{\mathbf{s}} = \mathbf{W}^\top \mathbf{x}$. |
| $\Sigma_{\mathbf{X}}$ | Covariance of $\mathbf{x}$ in used by the Pattern method. |
| $x_1, x_2, x_3$ | Features in the XOR example ($x_3$ is a suppressor variable). |
| $\mathcal{F}^+(\mathbf{x}_\star)$ | Ground-truth relevant pixels for an instance $\mathbf{x}_\star$. |

# B  Mathematical details

## B.1  Derivation of equation (7): Kernel-weighted ridge solution

We start with the PatternLocal definition from Eq. (6)

$$\mathbf{s}^{\text{PatternLocal}}(\mathbf{x}_\star) = \mathbf{a} = \arg\min_{\mathbf{u}\in\mathbb{R}^{D'}} \mathbb{E}_{\mathbf{x}\sim\mathcal{X}}\Big[\Pi_{\mathbf{x}_\star'}(\mathbf{h}_{\mathbf{x}_\star}(\mathbf{x}))\,\big\|\mathbf{h}_{\mathbf{x}_\star}(\mathbf{x}) - \mathbf{u}\,\tilde{y}\big\|_2^2\Big] + \lambda\,\|\mathbf{u}\|_2^2,$$

where the surrogate response is $\tilde{y} = \mathbf{w}^\top \mathbf{h}_{\mathbf{x}_\star}(\mathbf{x})$ and the kernel $\Pi_{\mathbf{x}_\star'} : \mathbb{R}^{D'} \to \mathbb{R}_{\geq 0}$ is normalized. For any function $f$ define

$$\mathbb{E}_{\Pi_{\mathbf{x}_\star'}}\big[f(\mathbf{x})\big] = \mathbb{E}_{\mathbf{x}\sim\mathcal{X}}\big[\Pi_{\mathbf{x}_\star'}(\mathbf{h}_{\mathbf{x}_\star}(\mathbf{x}))\,f(\mathbf{x})\big],$$

which coincides with the usual expectation because of the kernel's normalization. Using this notation, we set

$$J(\mathbf{u}) = \mathbb{E}_{\Pi_{\mathbf{x}_\star'}}\Big[\big\|\mathbf{h}_{\mathbf{x}_\star}(\mathbf{x}) - \mathbf{u}\,\tilde{y}\big\|_2^2\Big] + \lambda\,\|\mathbf{u}\|_2^2.$$

We can expand the squares like

$$\big\|\mathbf{h}_{\mathbf{x}_\star}(\mathbf{x}) - \mathbf{u}\,\tilde{y}\big\|_2^2 = \mathbf{h}_{\mathbf{x}_\star}(\mathbf{x})^\top \mathbf{h}_{\mathbf{x}_\star}(\mathbf{x}) - 2\,\tilde{y}\,\mathbf{h}_{\mathbf{x}_\star}(\mathbf{x})^\top \mathbf{u} + \tilde{y}^2\,\mathbf{u}^\top \mathbf{u},$$

and taking the kernel-weighted expectation gives

$$J(\mathbf{u}) = \mathbb{E}_{\Pi_{\mathbf{x}_\star'}}\big[\mathbf{h}_{\mathbf{x}_\star}(\mathbf{x})^\top \mathbf{h}_{\mathbf{x}_\star}(\mathbf{x})\big] - 2\,\mathbf{u}^\top\,\mathbb{E}_{\Pi_{\mathbf{x}_\star'}}\big[\tilde{y}\,\mathbf{h}_{\mathbf{x}_\star}(\mathbf{x})\big] + \mathbf{u}^\top \mathbf{u}\,\mathbb{E}_{\Pi_{\mathbf{x}_\star'}}\big[\tilde{y}^2\big] + \lambda\,\mathbf{u}^\top \mathbf{u}.$$

Then setting $\nabla_{\mathbf{u}} J(\mathbf{u}) = \mathbf{0}$ yields

$$-2\,\mathbb{E}_{\Pi_{\mathbf{x}_\star'}}\big[\tilde{y}\,\mathbf{h}_{\mathbf{x}_\star}(\mathbf{x})\big] + 2\,\big(\mathbb{E}_{\Pi_{\mathbf{x}_\star'}}[\tilde{y}^2] + \lambda\big)\,\mathbf{u} = \mathbf{0},$$

hence

$$\big(\mathbb{E}_{\Pi_{\mathbf{x}_\star'}}[\tilde{y}^2] + \lambda\big)\,\mathbf{u} = \mathbb{E}_{\Pi_{\mathbf{x}_\star'}}\big[\mathbf{h}_{\mathbf{x}_\star}(\mathbf{x})\,\tilde{y}\big].$$

Because $\mathbb{E}_{\Pi_{\mathbf{x}_\star'}}[\tilde{y}^2] + \lambda > 0$, the inverse exists and the ridge solution is

$$\mathbf{a}_{\ell_2} = \big(\mathbb{E}_{\Pi_{\mathbf{x}_\star'}}[\tilde{y}^2] + \lambda\big)^{-1}\,\mathbb{E}_{\Pi_{\mathbf{x}_\star'}}\big[\mathbf{h}_{\mathbf{x}_\star}(\mathbf{x})\,\tilde{y}\big].$$

Assuming that $\mathbf{h}_{\mathbf{x}_\star}(\mathbf{x})$ and $\tilde{y}$ is centered, the first-order moments vanish and we obtain the compact form

$$\mathbf{a}_{\ell_2} = \frac{\text{Cov}_{\Pi_{\mathbf{x}_\star'}}\big[\mathbf{h}_{\mathbf{x}_\star}(\mathbf{x}), \tilde{y}\big]}{\text{Var}_{\Pi_{\mathbf{x}_\star'}}[\tilde{y}] + \lambda}.$$

## B.2  Derivation for toy example

**Optimal classifier.**  The optimal classifier for the non-linear problem in Eq. (8) is given by

$$\tilde{f}(\mathbf{x}) = (x_1 - x_3)(x_2 + x_3),$$

as we can write

$$\tilde{f}(\mathbf{x}) = (x_1 - x_3)(x_2 + x_3) = ((m_1 + d) - d)((m_2 - d) + d) = m_1 m_2,$$

and we have $y = \text{sign}(m_1 m_2)$. However, the gradient of $\text{sign}(\tilde{f}(\mathbf{x}))$ is not well-defined when

$$\tilde{f}(\mathbf{x}) = 0 \quad\Leftrightarrow\quad (x_1 - x_3)(x_2 + x_3) = 0 \quad\Leftrightarrow\quad x_1 = x_3 \quad\text{or}\quad x_2 = -x_3.$$

These conditions define a union of two hyperplanes in $\mathbb{R}^3$ across which the sign function is discontinuous, and hence not differentiable. To address this, we instead use a smooth approximation to the sign function

$$f_\tau(\mathbf{x}) = \tanh\left(\frac{1}{\tau}\tilde{f}(\mathbf{x})\right), \qquad 0 < \tau \ll 1,$$

which is infinitely differentiable for all $\mathbf{x} \in \mathbb{R}^3$. As $\tau \to 0$, $f_\tau(\mathbf{x})$ converges pointwise to $\text{sign}(\tilde{f}(\mathbf{x}))$, while maintaining a smooth and well-defined gradient everywhere.

**Gradient.** To derive the local linear approximation of the smooth decision function, we apply the chain rule

$$\nabla f_\tau(\mathbf{x}) = \frac{1}{\tau} \operatorname{sech}^2\left(\frac{1}{\tau}\tilde{f}(\mathbf{x})\right) \cdot \nabla \tilde{f}(\mathbf{x}).$$

Expanding $\tilde{f}(\mathbf{x}) = x_1 x_2 + x_1 x_3 - x_2 x_3 - x_3^2$, we compute its gradient

$$\nabla \tilde{f}(\mathbf{x}) = \begin{bmatrix} x_2 + x_3 \\ x_1 - x_3 \\ x_1 - x_2 - 2x_3 \end{bmatrix}.$$

Evaluating at a point $\mathbf{x}_\star = (x_{1\star}, x_{2\star}, x_{3\star})$, we define

$$c_\star = \frac{1}{\tau} \operatorname{sech}^2\left(\frac{1}{\tau}\tilde{f}(\mathbf{x}_\star)\right),$$

and obtain the local weight vector

$$\mathbf{w} = \nabla f_\tau(\mathbf{x}_\star) = c_\star \begin{bmatrix} x_{2\star} + x_{3\star} \\ x_{1\star} - x_{3\star} \\ x_{1\star} - x_{2\star} - 2x_{3\star} \end{bmatrix}.$$

Thus, the first-order approximation becomes

$$f_\tau(\mathbf{x}) \approx f_\tau(\mathbf{x}_\star) + \mathbf{w}^\top (\mathbf{x} - \mathbf{x}_\star).$$

Finally, note that the identity $w_1 - w_2 + w_3 = 0$ holds.

**PatternLocal.** Let $\Pi_{\mathbf{x}_\star}(\mathbf{z}) = \varphi(\|\mathbf{z} - \mathbf{x}_\star\|)$ be an isotropic kernel and write $\mathbb{E}_{\Pi_{\mathbf{x}_\star}}[\cdot]$ for expectations taken with respect to the normalized kernel weights. The non–zero variances $\operatorname{Var}(m_1) = \operatorname{Var}(m_2) = 1$ and $\operatorname{Var}(d) = \sigma_d^2$ yield the full feature–covariance matrix

$$\Sigma_{\Pi_{\mathbf{x}_\star}} = \operatorname{Cov}_{\Pi_{\mathbf{x}_\star}}[\mathbf{x}, \mathbf{x}] = \begin{pmatrix} 1 + \sigma_d^2 & -\sigma_d^2 & \sigma_d^2 \\ -\sigma_d^2 & 1 + \sigma_d^2 & -\sigma_d^2 \\ \sigma_d^2 & -\sigma_d^2 & \sigma_d^2 \end{pmatrix}.$$

With identity mapping $h_{\mathbf{x}_\star}(\mathbf{x}) = \mathbf{x}$ and $\lambda = 0$, the PatternLocal explanation is

$$\mathbf{a} = \frac{\operatorname{Cov}_{\Pi_{\mathbf{x}_\star}}[\mathbf{x}, \tilde{y}]}{\operatorname{Var}_{\Pi_{\mathbf{x}_\star}}[\tilde{y}]}, \qquad \tilde{y} = \mathbf{w}^\top \mathbf{x}.$$

Using the gradient weights $\mathbf{w} = (w_1, w_2, w_3)^\top$ of Eq. (9) in the main text, obeying $w_1 - w_2 + w_3 = 0$,

$$\operatorname{Cov}_{\Pi_{\mathbf{x}_\star}}[\mathbf{x}, \tilde{y}] = \Sigma_{\Pi_{\mathbf{x}_\star}} \mathbf{w} = \begin{pmatrix} (1 + \sigma_d^2)w_1 - \sigma_d^2 w_2 + \sigma_d^2 w_3 \\ -\sigma_d^2 w_1 + (1 + \sigma_d^2)w_2 - \sigma_d^2 w_3 \\ \sigma_d^2 w_1 - \sigma_d^2 w_2 + \sigma_d^2 w_3 \end{pmatrix}.$$

The third component simplifies immediately,

$$\operatorname{Cov}_\Pi[x_3, \tilde{y}] = \sigma_d^2(w_1 - w_2 + w_3) = 0,$$

so $a_3 = 0$. For the remaining two components, use the same identity once to obtain

$$\operatorname{Cov}_\Pi[x_1, \tilde{y}] = w_1, \qquad \operatorname{Cov}_\Pi[x_2, \tilde{y}] = w_2.$$

Consequently

$$\mathbf{a} \propto \begin{pmatrix} w_1 \\ w_2 \\ 0 \end{pmatrix},$$

which removes the suppressor variable $x_3$ even though $w_3 \neq 0$.

## B.3 Sample complexity of kernel-weighted moment estimation

It is important to note that an explicit matrix covariance estimate (and its multiplication) is not always required, especially when the surrogate already contains an inverse-covariance form. The key computation in PatternLocal is the kernel-weighted moment

$$a_{\ell_2} = \frac{\mathrm{Cov}_{\Pi_{\mathbf{x}'_\star}}\big[\mathbf{h}_{\mathbf{x}_\star}(\mathbf{x}), \tilde{y}\big]}{\mathrm{Var}_{\Pi_{\mathbf{x}'_\star}}[\tilde{y}] + \lambda}, \qquad \tilde{y} = \mathbf{w}^\top \mathbf{h}_{\mathbf{x}_\star}(\mathbf{x}),$$

which for binary classification requires only (1) the scalar local variance of $\tilde{y}$ and (2) the vector cross-covariance with the simplified features $\mathbf{h}_{\mathbf{x}_\star}(\mathbf{x})$. Both moments are estimated with a Nadaraya–Watson kernel smoother. For any function $g(\mathbf{x})$ we write

$$\widehat{\mathbb{E}}[g] = \frac{\sum_{i=1}^{n_{\mathrm{loc}}} K_\sigma\big(\|\mathbf{h}_{\mathbf{x}_\star}(\mathbf{x}_i) - \mathbf{0}\|\big)\, g(\mathbf{x}_i)}{\sum_{i=1}^{n_{\mathrm{loc}}} K_\sigma\big(\|\mathbf{h}_{\mathbf{x}_\star}(\mathbf{x}_i) - \mathbf{0}\|\big)},$$

where $K_\sigma$ is an isotropic kernel with bandwidth $\sigma$ in the $D'$-dimensional interpretable space, and $n_{\mathrm{loc}}$ is the number of local samples.

Classical Nadaraya–Watson analysis [36] gives for each entry of the conditional covariance

$$\mathrm{bias} = O(\sigma^2), \qquad \mathrm{var} = O\big((n_{\mathrm{loc}}\sigma^{D'})^{-1}\big).$$

Balancing these two terms yields the optimal bandwidth $\sigma^\star \propto n_{\mathrm{loc}}^{-1/(4+D')}$ and the corresponding mean-squared error

$$\mathrm{MSE} = O\big(n_{\mathrm{loc}}^{-4/(4+D')}\big).$$

Equivalently, to achieve an $\varepsilon$-accurate estimate of the required local moments, understood as $\mathrm{RMSE} \le \varepsilon$, one needs

$$n_{\mathrm{loc}} = \Omega\big(\varepsilon^{-(4+D')/2}\big),$$

which scales only with the *simplified* input dimension $D' \ll D$.

When $D'$ is not small compared to $n_{\mathrm{loc}}$, we apply Ledoit–Wolf shrinkage to the local covariance. Ledoit and Wolf [16] analyze a linear shrinkage estimator that remains invertible even when $D' > n_{\mathrm{loc}}$, is well-conditioned in probability, and is asymptotically optimal when $D'/n_{\mathrm{loc}}$ stays bounded. Importantly, PatternLocal does not require a matrix inverse, since Eq. (7) divides only by the *scalar* variance of $\tilde{y}$.

Finally, recent locality-aware sampling results for GLIME [31] provide sample-complexity bounds that show exponentially faster convergence and substantially fewer samples than LIME under sub-Gaussian local sampling and regularization. PatternLocal can adopt the same sampling strategy, which further reduces the number of data points per explanation without altering its closed-form update.

# C Dataset generation

## C.1 XAI-TRIS Benchmark

Following the methodology outlined in Clark et al. [8], the XAI-TRIS benchmark dataset consists of images of size $64 \times 64$, represented as $\mathcal{D} = \{(\mathbf{x}^{(n)}, y^{(n)})\}_{n=1}^{N}$. The feature dimensionality is $D = 64^2 = 4\,096$, and the dataset contains $N = 40\,000$ samples. These samples are independent and identically distributed (i.i.d.) realizations of random variables $\mathbf{X}$ and $Y$, governed by the joint probability density function $P_{\mathbf{X},Y}(\mathbf{x}, y)$.

Each instance $\mathbf{x}^{(n)}$ comprises a *foreground* signal $\mathbf{a}^{(n)} \in \mathbb{R}^D$, which is class-dependent and defines the ground truth, combined with *background* noise $\boldsymbol{\eta}^{(n)} \in \mathbb{R}^D$. The additive generation process is defined as

$$\mathbf{x}^{(n)} = \alpha(R^{(n)} \circ (H \circ \mathbf{a}^{(n)})) + (1 - \alpha)(G \circ \boldsymbol{\eta}^{(n)}), \tag{12}$$

where $\alpha$ determines the relative contribution of signal and noise. The signal $\mathbf{a}^{(n)}$ is based on tetromino shapes that depend on the binary class label $y^{(n)} \sim \text{Bernoulli}\left(\frac{1}{2}\right)$. To smooth the signal, a Gaussian spatial filter $H : \mathbb{R}^D \to \mathbb{R}^D$ with a standard deviation $\sigma_{\text{smooth}} = 1.5$ is applied, using a maximum support threshold of $5\%$.

The background noise $\eta^{(n)}$ is sampled as $\eta^{(n)} \sim \mathcal{N}(0, \mathbf{I}_D)$. Two noise scenarios are considered:

1. **WHITE**: The operator $G$ is the identity function, resulting in uncorrelated noise.
2. **CORR**: The operator $G : \mathbb{R}^D \to \mathbb{R}^D$ is a Gaussian spatial filter with $\sigma_{\text{smooth}} = 10$, introducing spatially correlated noise.

We analyze three distinct binary classification scenarios:

1. **LIN**: The operator $R^{(n)}$ is the identity operator. The signal $\mathbf{a}^{(n)}$ is defined as a 'T'-shaped tetromino $\mathbf{a}^{\text{T}}$ located near the top-left if $y = 0$, and as an 'L'-shaped tetromino $\mathbf{a}^{\text{L}}$ located near the bottom-right if $y = 1$.

2. **XOR**: The operator $R^{(n)}$ is the identity operator, and each instance contains both $\mathbf{a}^{\text{T}}$ and $\mathbf{a}^{\text{L}}$. For $y = 0$, the signals are $\mathbf{a}^{\text{XOR}++} = \mathbf{a}^{\text{T}} + \mathbf{a}^{\text{L}}$ and $\mathbf{a}^{\text{XOR}-} = -\mathbf{a}^{\text{T}} - \mathbf{a}^{\text{L}}$. For $y = 1$, the signals are $\mathbf{a}^{\text{XOR}+-} = \mathbf{a}^{\text{T}} - \mathbf{a}^{\text{L}}$ and $\mathbf{a}^{\text{XOR}-+} = -\mathbf{a}^{\text{T}} + \mathbf{a}^{\text{L}}$.

3. **RIGID**: The operator $R^{(n)}$ applies a rigid-body transformation. In this case, the tetromino shapes $\mathbf{a}^{\text{T}}$ and $\mathbf{a}^{\text{L}}$ are randomly translated and rotated in increments of $90°$.

Lastly, the transformed signal and noise components are horizontally concatenated into matrices and normalized by the Frobenius norm and a weighted sum is calculated with the scalar $\alpha \in [0, 1]$ which determines the signal-to-noise ratio (SNR).

This results in six scenarios across $64 \times 64$ image sizes. The ground truth set, representing the important pixels based on the positions of the tetromino shapes, is

$$\mathcal{F}^+(\mathbf{x}^{(n)}) := \{\, d \in \{1, \dots, D\} : (R^{(n)} \circ (H \circ \mathbf{a}^{(n)}))_d \neq 0\}. \tag{13}$$

For the LIN scenario, the presence of $\mathbf{a}^{\text{T}}$ is as informative as the absence of $\mathbf{a}^{\text{L}}$, and vice versa. Therefore, the set of important pixels for this setting is

$$\mathcal{F}^+(\mathbf{x}^{(n)}) := \{\, d \in \{1, \dots, D\} : (H \circ \mathbf{a}^{\text{T}}_d \neq 0) \ \vee \ (H \circ \mathbf{a}^{\text{L}}_d \neq 0)\}. \tag{14}$$

The ground truth $\mathcal{F}^+(\mathbf{x}^{(n)})$ defines important pixels based on the data generation process. However, a model may rely on only a subset of these for its prediction, which is equally valid. As in Clark et al. [8], we employ metrics to de-emphasize the impact of false-negative omissions.

For model training and later analysis, each dataset is split into three subsets $\mathcal{D}_{\text{train}}$, $\mathcal{D}_{\text{val}}$, and $\mathcal{D}_{\text{test}}$ with a 90/5/5 ratio.

**XAI-TRIS Benchmark for Ablation Study**    Additionally, we generate another XAI-TRIS benchmark dataset following the same procedure but for $D = 8^2 = 64$ and $N = 10\,000$. This is only used for the ablation study in Appendix G.

## C.2 Artificial Lesion MRI Benchmark

The methodology of constructing the Artificial Lesion MRI Benchmark dataset is described in Oliveira et al. [19] and we provide a brief overview here. See Figure 7 for examples of instances.

**Source images.** They selected $1\,007$ T1–weighted axial volumes of healthy adults (22–37 y) from the Human Connectome Project (HCP). After the standard FSL/FREESURFER "minimal" preprocessing and defacing pipelines, each slice with background occupancy $< 55\%$ was retained. Raw matrices ($260 \times 311$) were zero-padded vertically, centre-cropped horizontally to $270 \times 270$, clipped to the intensity range $[0, 0.7]$, and bicubic down-sampled to $128 \times 128$ so as to match the resolution used in the main paper.

**Lesion synthesis.** *(i) Prototype masks.* White Gaussian noise ($256 \times 256$) was smoothed with a $\sigma = 2$ kernel, Otsu-thresholded, and morphologically processed (erosion–opening–erosion) to yield irregular blobs. *(ii) Shape selection.* From every connected component they computed the *compactness*

$$c = \frac{4\pi A}{p^2}, \qquad c \in (0, 1],$$

where $A$ is area and $p$ its perimeter. Components with $c \geq 0.8$ were labelled **regular**, those with $c \leq 0.4$ **irregular**. *(iii) Boundary refinement.* Candidate shapes were zero-padded by two pixels and softened with a $\sigma = 0.75$ Gaussian blur, yielding final binary masks $L \in \{0, 1\}^{256 \times 256}$.

**Image composition.** For every background slice $B$: *(i)* Draw $k \sim \mathcal{U}\{3, 4, 5\}$ lesions $L_{1:k}$ of the same shape class. *(ii)* Uniformly translate each $L_j$ inside the brain bounding box, rejecting placements with overlap (IOU$> 0$). *(iii)* Aggregate the lesion mask $L(x, y) = \max_j L_j(x, y)$ and mix intensities

$$X(x, y) = B(x, y)\,[1 - L(x, y)] + \alpha\, L(x, y),$$

fixing the signal-to-noise weighting $\alpha = 0.5$ for all experiments. Pixels with $L(x, y) = 1$ constitute the exact attribution ground truth supplied to evaluation metrics.

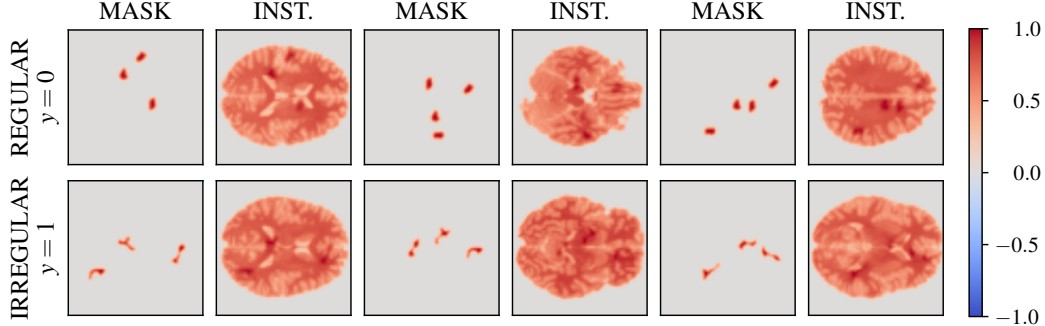

Figure 7: Example of the Artificial Lesion MRI Benchmark dataset. Each pair of panels shows the lesion mask *(MASK)* and the corresponding synthetic T1-weighted slice after intensity blending *(INST.)*. Three independent examples are displayed for each of the two lesion-shape classes: compact *regular* lesions (top row, $y = 0$) and low-compactness *irregular* lesions (bottom row, $y = 1$). All slices are $128 \times 128$ pixels.

# D Experimental details

## D.1 Model architectures and training

All classifiers are implemented in `PyTorch 2.6` and `PyTorch-Lightning 2.5`. Training, early stopping, and checkpointing are handled entirely by the `Lightning` trainer; the relevant source code is in the GitHub repository for full transparency. Unless stated otherwise, *all* models are trained with the same optimization hyperparameters listed in Table 2.

**Multi-layer perceptron (MLP).** The network is a 4-layer perceptron that receives a flattened input vector. Hidden dimensions are $\{128, 64, 32, 16\}$ with ReLU activations, batch–normalization after every linear layer, and 25% drop-out regularization. The output layer is fully connected with $|\mathcal{Y}|$ logits.

**Convolutional neural network (CNN).** The canonical CNN processes single-channel square images of side $64 \times 64$ or $128 \times 128$ pixels, selected by the `input_size` argument (4096, 16384 vector elements, respectively). It stacks four convolution blocks

$$\text{Conv}(1{\rightarrow}4, k{=}4) \rightarrow \text{Conv}(4{\rightarrow}8, k{=}4) \rightarrow \text{Conv}(8{\rightarrow}16, k{=}4) \rightarrow \text{Conv}(16{\rightarrow}32, k{=}4),$$

each followed by ReLU and a $2{\times}2$ max-pool, after which the feature map is flattened and passed to a single fully connected output layer. When the input is $8{\times}8$, we employ a compact architecture with two $3{\times}3$ convolutions (channels $1{\rightarrow}8{\rightarrow}16$), each followed by batch-norm, ReLU, max-pool, and 25% drop-out. The resulting $4{\times}4$ feature map feeds a single fully connected classifier.

**Optimisation and early stopping.** Parameters are updated with Adam ($\eta_0 = 1 \times 10^{-4}$). A `ReduceLROnPlateau` scheduler monitors validation accuracy and scales the learning rate by $0.1$ after 100 epochs. Training terminates via early stopping under the same patience condition, and the model state with the *highest* validation accuracy is restored before final testing. Mini-batches contain 128 samples, and the hard cap on epochs is 500, although the best model is usually after around $\sim$60–120 epochs.

Table 2: Hyperparameters used for model training.

| Hyperparameter | Value |
|---|---|
| Initial learning rate | $1 \times 10^{-4}$ |
| Batch size | 128 |
| LR-scheduler factor | 0.1 |
| Patience (LR + early stop) | 100 epochs |
| Maximum training epochs | 500 |
| Optimiser | Adam |
| Loss function | Cross-entropy |

All random seeds are fixed through `pl.seed_everything` to ensure that the reported numbers are exactly reproducible. For every model used, the test accuracy exceeds 90 %.

## D.2 Hyperparameter optimization

**Optimization procedure.** We perform Bayesian optimization with the Tree-of-Parzen Estimators (TPE) algorithm [4] as implemented in the `hyperopt` package [5]. Each run consists of 200 trials; during every trial, the candidate configuration is evaluated on the validation set, and the mean Earth Mover's Distance (EMD) between the normalized explanation and the ground-truth mask is minimized.

**Locally linear methods.** LIME and PatternLocal are tuned independently. PatternLocal optimizes the three parameters listed in Table 3 and the LIME parameters, yielding four variables in total.

Table 3: Hyperparameter search spaces explored by the Bayesian optimizer. A dash ($-$) indicates that no hyperparameters were tuned. For the $128 \times 128$ images, the upper bound of the bandwidth for LIME and PatternLocal is increased.

| Method | Hyperparameters (range) |
|---|---|
| LIME | Bandwidth $\in [0.5, 30.0]$ |
| PatternLocal | Bandwidth $\in [0.5, 30.0]$, Regularization $\lambda \in [10^{-5}, 10^2]$
Kernel $\in \{\texttt{gaussian}, \texttt{epanechnikov}\}$ |
| Laplace | $-$ |
| Sobel | $-$ |
| Saliency | $-$ |
| GuidedBackprop | $-$ |
| DeepLift | $-$ |
| Integrated Gradients | $n_{\text{steps}} \in [10, 200]$, Method $\in \{\texttt{riemann\_trapezoid}, \texttt{gausslegendre}\}$ |
| GradientShap | $n_{\text{samples}} \in [5, 50]$, $\sigma_{\text{noise}} \in [0.00, 0.30]$ |

**Baseline methods.** We run with their default implementations for filter-based methods (Laplace, Sobel) and several XAI methods (Saliency, DeepLift, GuidedBackprop). Integrated Gradients and GradientShap have modest search spaces that cover the step count, integration scheme, and, for GradientShap, the noise characteristics.

### D.3 Computational details

**Hardware.** All experiments were executed on a local high-performance computing (HPC) cluster equipped with **Intel® Xeon E5-2650 v4** CPUs (12 cores, 24 threads, 2.20 GHz) and **256 GB** RAM per node. No dedicated GPUs were required. Jobs were managed with SLURM 22.05 and ran under AlmaLinux 9.5.

**Software.** The codebase is primarily written in Python 3.13.0. Key libraries are:

- NumPy 2.1.3,
- PyTorch 2.6 for model definition,
- PyTorch-Lightning 2.5 for model training and evaluation,
- scikit-learn 1.6.1 for classical baselines and metrics,
- hyperopt 0.2.7 for Bayesian optimization,
- POT 0.9.5 for Earth-Mover-Distance evaluation,
- Hydra 1.3.2 for experiment handling.

The requirements.txt file includes exact versions and is included in the project repository.

**Runtime for classifiers.** Training a single classifier on *XAI-TRIS* or the Artificial-Lesion MRI data required ∼10-15 min on one CPU core. By far the dominant cost stemmed from hyperparameter optimization of the XAI methods. For each combination of *scenario* (LIN, XOR, RIGID) and signal-to-noise ratio ($\alpha \in \{0.1, \ldots, 0.90\}$) we ran 200 TPE iterations, evaluating on the full validation split (500–2000 instances, depending on the dataset). After ∼100-150 iterations, we confirmed the convergence plateaus. Each optimization run was parallelized across 12 CPU cores via multiprocessing, resulting in a wall-clock time of 12-24 hours.

**Runtime for XAI methods.** We report runtime measurements for the XAI methods in Table 4. PatternLocal adds a single weighted regression per explanation on top of the cost of the local surrogate. Since fitting the surrogate (e.g., with LIME) is the most expensive step, the additional regression keeps the runtime in the same order of magnitude. Moreover, PatternLocal only regresses on samples in the local neighborhood, and in practice, we further reduce cost by discarding samples with very low weight. Like LIME, we use superpixels to reduce computational cost and include a low-rank approximation. See more details in Appendix E.1.

Table 4: Runtime of XAI methods.

| XAI method | Avg. runtime $\pm$ std (s) |
|---|---|
| PatternLocal | $12.9 \pm 1.5$ |
| LIME | $7.4 \pm 0.3$ |
| PatternLocal + Superpixel | $3.6 \pm 0.3$ |
| LIME + Superpixel | $1.4 \pm 0.1$ |
| PatternLocal + LowRank | $7.7 \pm 0.4$ |
| LIME + LowRank | $6.9 \pm 0.2$ |

Although runtimes are difficult to compare across different hardware and implementations, we control for this by keeping all settings fixed and averaging over 1000 instances of the XAI-TRIS dataset on an AMD EPYC 9124 16c/32t 3.0 GHz machine (also for LIME inference). Given that PatternLocal is only slightly slower than other model-agnostic methods such as LIME or KernelSHAP, while consistently achieving better explanations, we believe the overhead is justified.

# E   Extended results

## E.1   XAI-TRIS Benchmark

**Quantitative evaluation.**   In Figure 8, we investigates how different choices of $\mathbf{h}_{\mathbf{x}_\star}(\cdot)$ and $Q(\cdot) = \|\cdot\|_1$ affect explanation quality. We compare PatternLocal and LIME with a low-rank representation and superpixel representations.

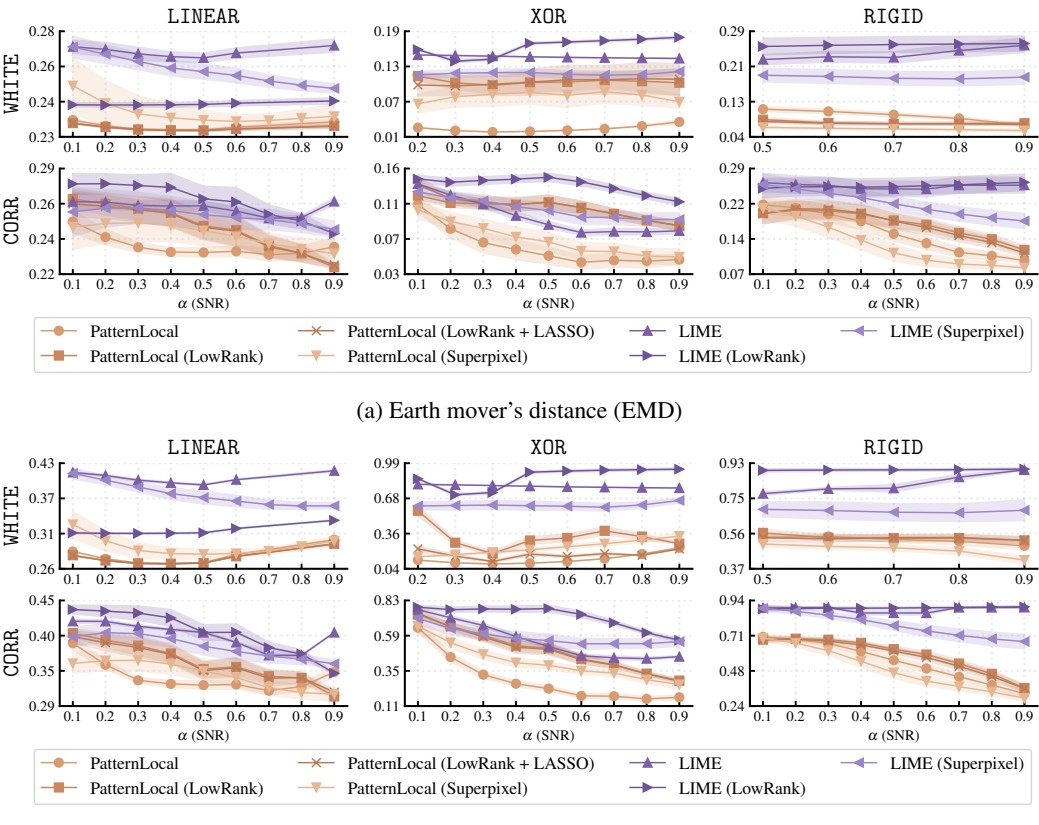

(a) Earth mover's distance (EMD)

(b) Importance mass error (IME)

Figure 8: Quantitative evaluation of feature importance maps for the MLP model using $64 \times 64$ XAI-TRIS benchmark images generated by variants of the PatternLocal and LIME method, as a function of the SNR. The MLP model achieves at least $90\%$ accuracy, and method hyperparameters were optimized for EMD. The top panel (8a) shows results based on EMD, while the bottom panel (8b) presents performance in terms of IME. Shaded regions indicate standard deviation as the standard error is too small to be visible.

**Remark.** Not all curves decrease with $\alpha$, which can be explained by two effects. For edge-detector baselines such as Sobel and Laplace, increasing $\alpha$ sharpens tetromino boundaries and causes the filters to respond outside the true region, increasing the EMD/IME. For local surrogate methods such as LIME and PatternLocal, higher $\alpha$ improves the signal-to-noise ratio and often leads to better local fits. However, the spike around $\alpha \approx 0.9$ on WHITE-LINEAR is likely numerical: as shown in Figure 5, the dataset becomes rank-deficient at high $\alpha$, making the regression unstable.

Table 5: Empirical matrix rank as a function of contrast $\alpha$. The drop to rank 2 explains the spike for LINEAR WHITE, while RIGID WHITE remains full-rank longer, resulting in smoother curves.

<table>
<tr><td colspan="6" align="center">(a) LINEAR WHITE</td></tr>
<tr><td>$\alpha$</td><td>0.5</td><td>0.6</td><td>0.7</td><td>0.8</td><td>0.9</td></tr>
<tr><td>rank</td><td>4096</td><td>4096</td><td>4096</td><td>3554</td><td>**2**</td></tr>
</table>

<table>
<tr><td colspan="6" align="center">(b) RIGID WHITE</td></tr>
<tr><td>$\alpha$</td><td>0.5</td><td>0.6</td><td>0.7</td><td>0.8</td><td>0.9</td></tr>
<tr><td>rank</td><td>4096</td><td>4096</td><td>4096</td><td>4096</td><td>2881</td></tr>
</table>

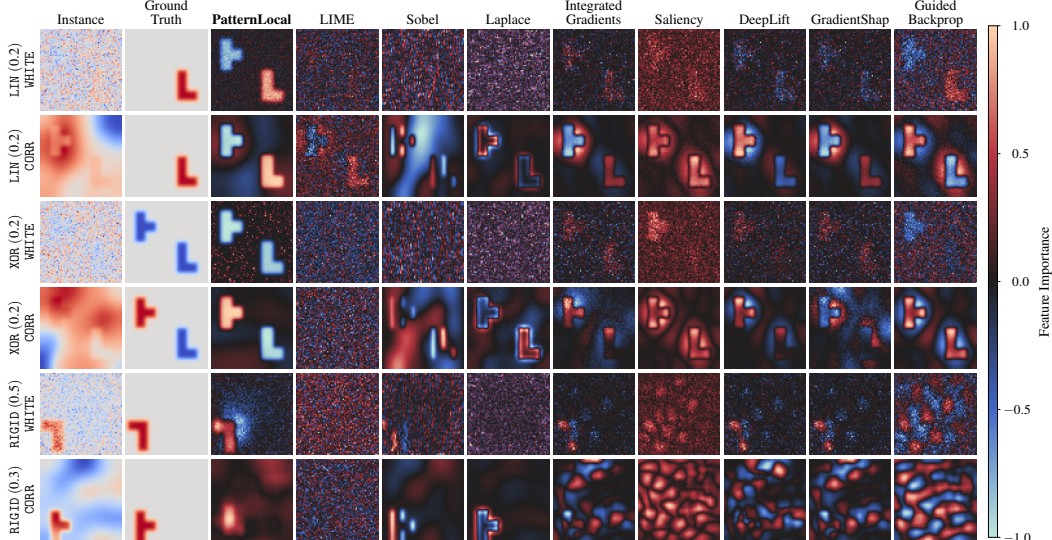

Figure 9: Qualitative comparison on XAI-TRIS for the MLP model, $\mathbf{h}_{\mathbf{x}_\star}(\cdot)$ being the identity function and $Q(\cdot) = \|\cdot\|_1$. Each row shows one of the six benchmark configurations: LIN WHITE ($\alpha = 0.2$), LIN CORR ($\alpha = 0.2$), XOR WHITE ($\alpha = 0.2$), XOR CORR ($\alpha = 0.2$), RIGID WHITE ($\alpha = 0.5$), and RIGID CORR ($\alpha = 0.3$). Columns display the input *Instance*, the *Ground Truth* attribution mask, our *PatternLocal*, and nine baselines (LIME, Sobel, Laplace, IntegratedGradients, Saliency, DeepLift, GradientShap, GuidedBackprop). PatternLocal aligns closest with the ground truth across all scenarios, whereas filter-based methods only succeed in the RIGID CORR case.

**Qualitative evaluation.**    In Figure 9, we show qualitative examples of explanations on the XAI-TRIS Benchmark dataset obtained with the MLP model. For PatternLocal and LIME we set $\mathbf{h}_{\mathbf{x}_\star}(\cdot)$ to the identity and use the squared $\ell_2$–penalty, $R(\cdot) = Q(\cdot) = \|\cdot\|_2^2$. It is evident that PatternLocal produces better explanations, most markedly in the WHITE *scenarios* and consistently better than LIME in every setting. As expected, the filter-based methods (Sobel, Laplace) perform competitively only in the RIGID CORR case.

Figure 10 shows examples of explanation for different choices of $\mathbf{h}_{\mathbf{x}_\star}(\cdot)$ and $Q(\cdot) = \|\cdot\|_1$. We compare PatternLocal and LIME with a low-rank representation and superpixel representations. We use the naming PatternLocal ($\mathbf{h}_{\mathbf{x}_\star}(\cdot) = \text{id}, Q(\cdot) = \|\cdot\|_2^2$), PatternLocal LowRank ($\mathbf{h}_{\mathbf{x}_\star}(\cdot) = \text{lowrank}$, $Q(\cdot) = \|\cdot\|_2^2$), and PatternLocal LowRank LASSO ($\mathbf{h}_{\mathbf{x}_\star}(\cdot) = \text{superpixel}$ and $Q(\cdot) = \|\cdot\|_1$).

### E.2    Artificial Lesion MRI Benchmark

The **Artificial Lesion MRI Benchmark** dataset [19] was created to provide a semi–realistic, fully controlled setting for evaluating XAI methods on MRI-based lesion detection. Each image is a down-sampled axial slice of size $128 \times 128$ containing one or more synthetically inserted lesions. The input dimensionality is highly imbalanced ($N = 2\,500 < D = 16\,384$). This pronounced $D > N$ regime makes local surrogate approaches such as LIME, KernelSHAP, and our PatternLocal particularly sensitive to the choice of the input–simplification mapping $\mathbf{h}_{\mathbf{x}_\star}(\cdot)$ and to regularization. In all experiments, we therefore segment each slice into superpixels with slic before fitting the surrogates, and set both regularizers to the squared $\ell_2$ norm, $R(\cdot) = Q(\cdot) = \|\cdot\|_2^2$. The classifier is a CNN model achieving $> 90\%$ accuracy on the test set.

**Quantitative evaluation.**    Table 6 reports the mean Earth Mover's Distance (EMD) and Importance Mass Error (IME) over the entire test set. PatternLocal and LIME underperform relative to the other XAI methods. We suspect that this is primarily because the latter suppresses many low-level attributions that inflate the error of the PatternLocal and LIME methods. Crucially, this strategy considers every superpixel, even those that neither method deems highly relevant. It is not uncommon to perform feature selection or only consider the most important superpixels. A more realistic assessment is obtained by considering only superpixels whose importance exceeds a high threshold.

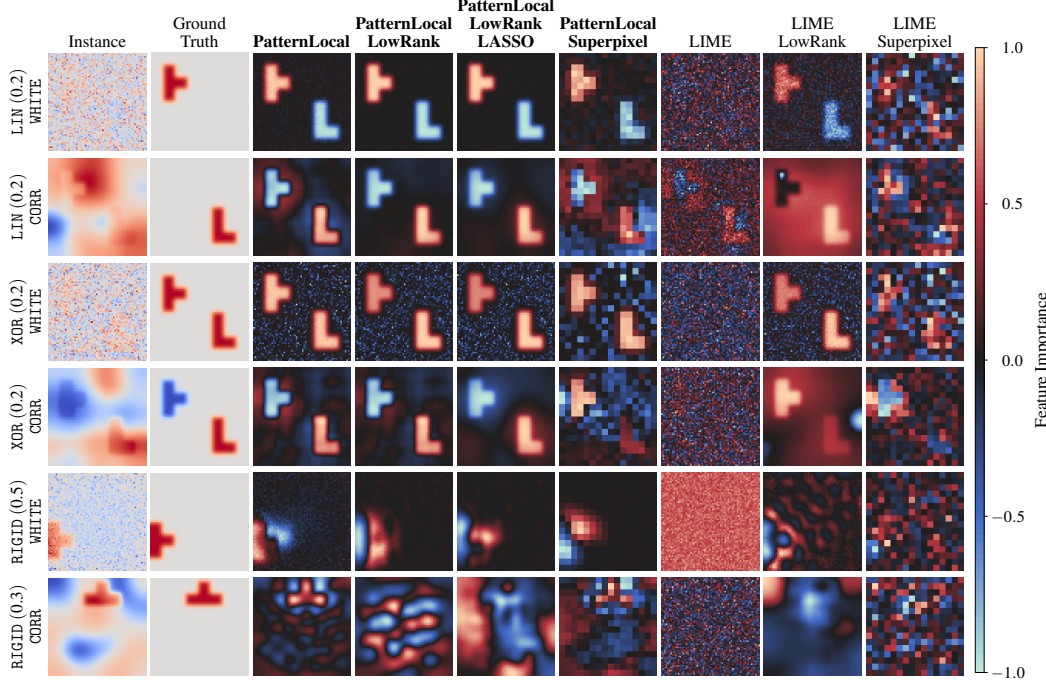

Figure 10: Qualitative comparison on XAI-TRIS for the MLP model for various choices of $\mathbf{h}_{\mathbf{x}_*}(\cdot)$ and $Q(\cdot) = \|\cdot\|_1$. Each row shows one of the six benchmark configurations: LIN WHITE ($\alpha = 0.2$), LIN CORR ($\alpha = 0.2$), XOR WHITE ($\alpha = 0.2$), XOR CORR ($\alpha = 0.2$), RIGID WHITE ($\alpha = 0.5$), and RIGID CORR ($\alpha = 0.3$). Columns display the input *Instance*, the *Ground Truth* attribution mask, and four variants of PatternLocal and three variants of LIME. PatternLocal is robust across representations, degrading only in the most challenging RIGID CORR setting when a low-rank approximation is used. LIME benefits markedly from the low-rank approximation but deteriorates for superpixels.

Restricting the comparison to above 0.9 (Table 7) changes the ranking: PatternLocal now attains the lowest EMD and a markedly reduced IME, whereas LIME and the edge filters deteriorate. The gradient-based methods improve their IME scores but exhibit extreme variance, indicating that their few high-magnitude attributions are positioned correctly only about half the time. Additionally, on average, PatternLocal selects $2.5 \pm 1.2$ superpixels per slice vs. $4.5 \pm 2.1$ for LIME. The contrast between the two evaluation protocols highlights a well-known drawback of superpixel explanations: much of the attribution mass is distributed over moderately important segments that are irrelevant for human interpretation yet heavily penalize set-wise metrics such as EMD and IME. By focusing on high-confidence regions, PatternLocal exposes its intended advantage in reducing false-positive attributions.

Table 6: Comparison on the Artificial Lesion MRI benchmark (*all* features).

| Method | EMD | IME |
|---|---|---|
| PatternLocal (superpixel) | $0.166 \pm 0.034$ | $0.969 \pm 0.016$ |
| LIME (superpixel) | $0.171 \pm 0.025$ | $0.971 \pm 0.009$ |
| Laplace | $0.131 \pm 0.022$ | $0.956 \pm 0.011$ |
| Sobel | $0.137 \pm 0.022$ | $0.950 \pm 0.011$ |
| Integrated Gradients | $0.101 \pm 0.021$ | $0.894 \pm 0.026$ |
| Saliency | $0.124 \pm 0.022$ | $0.933 \pm 0.018$ |
| DeepLift | $0.103 \pm 0.021$ | $0.897 \pm 0.026$ |
| GradientShap | $0.102 \pm 0.021$ | $0.898 \pm 0.025$ |
| Guided Backprop | $0.124 \pm 0.021$ | $0.933 \pm 0.018$ |

Table 7: Comparison on the Artificial Lesion MRI benchmark (top features).

| Method | EMD | IME |
|---|---|---|
| PatternLocal (superpixel) | $0.102 \pm 0.065$ | $0.827 \pm 0.057$ |
| LIME (superpixel) | $0.196 \pm 0.062$ | $0.976 \pm 0.077$ |
| Laplace | $0.185 \pm 0.056$ | $1.000 \pm 0.008$ |
| Sobel | $0.276 \pm 0.051$ | $1.000 \pm 0.003$ |
| Integrated Gradients | $0.166 \pm 0.038$ | $0.538 \pm 0.428$ |
| Saliency | $0.171 \pm 0.048$ | $0.804 \pm 0.331$ |
| DeepLift | $0.171 \pm 0.047$ | $0.569 \pm 0.422$ |
| GradientShap | $0.158 \pm 0.051$ | $0.554 \pm 0.422$ |
| Guided Backprop | $0.167 \pm 0.049$ | $0.803 \pm 0.336$ |

**Qualitative evaluation.** Explanations are shown in Figure 11. Across all slices, PatternLocal highlights fewer superpixels, and the ones it does highlight align more closely with the ground-truth lesions than those produced by LIME. Both methods are nonetheless limited by the superpixel resolution: lesions that cross superpixel boundaries cannot be perfectly recovered. Filter-based methods (Sobel, Laplace) fail completely because the lesions are diffuse and lack strong boundaries; XAI methods (Integrated Gradients, Saliency, DeepLift, GradientShap, GuidedBackprop) tend to produce noisy attribution maps with only occasional alignment to the lesions.

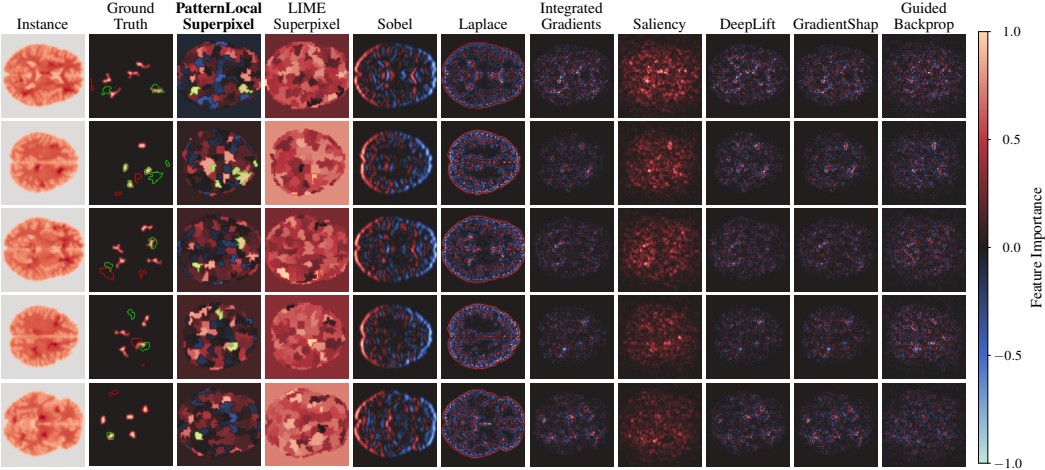

Figure 11: Examples from the artificial lesion MRI dataset, showing 5 MRI slices (rows). The first column displays the original MRI slices with artificially generated lesions; the second column (Ground Truth) shows the true lesion locations. The subsequent columns show feature importance maps from PatternLocal, LIME, Sobel, Laplace, IntegratedGradients, Saliency, DeepLift, GradientShap, and GuidedBackprop. We use a CNN model, $\mathbf{h}_{\mathbf{x}_\star}(\cdot)$, with superpixels, and set $R(\cdot) = Q(\cdot) = \| \cdot \|_2^2$ for both PatternLocal and LIME. Overlaid on the ground truth are superpixels with feature importance above 0.9 from PatternLocal (green) and LIME (red). PatternLocal explanations appear better aligned with the lesion locations.

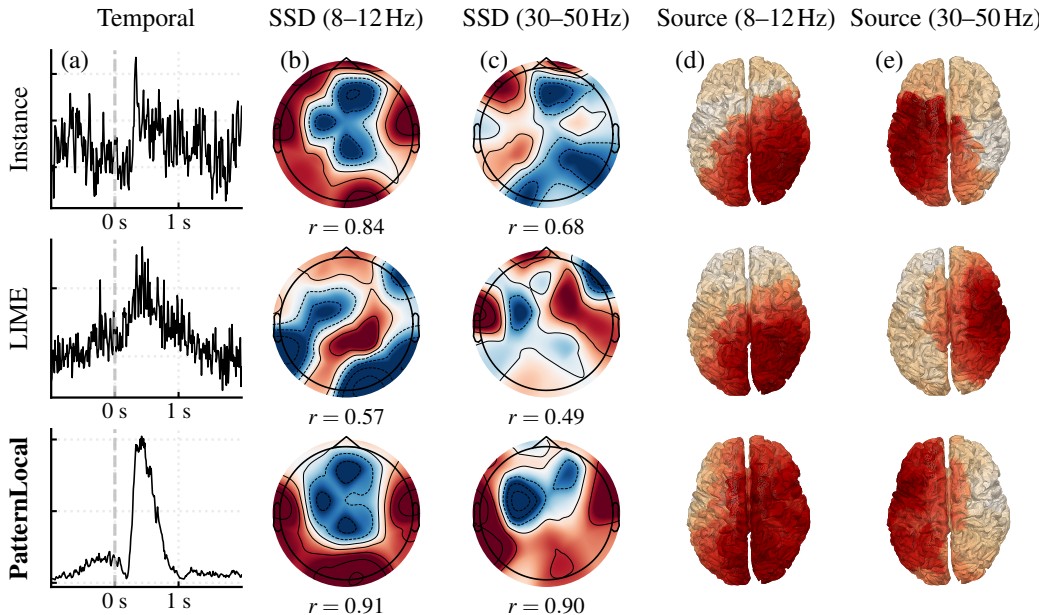

Figure 12: Physiological validation of EEG explanations for a representative right-hand motor imagery trial. (a) Channel-averaged explanation signals for a single instance, LIME, and PatternLocal. PatternLocal shows a sharp, cue-locked increase around stimulus onset, whereas LIME produces a broader and less distinct response. (b–c) Spatio-Spectral Decomposition (SSD) topographies derived from the explanation signals in the alpha (8–12 Hz) and gamma (30–50 Hz) ranges, compared with SSD patterns from the raw EEG. The $r$ values denote the goodness of fit of a single dipole to each pattern; PatternLocal yields the most consistent and physiologically plausible fits across both frequency ranges. (d–e) Source estimates obtained with eLORETA from SSD-filtered explanation signals show activation over peri-rolandic cortex with stronger responses contralateral to the imagined hand. Overall, PatternLocal provides spatial patterns and source activity that better reflect known motor-imagery physiology than LIME under matched conditions.

### E.3  EEG Motor Imagery dataset

Figure 12 illustrates a trial from the EEG motor imagery experiment used for physiological validation. It shows how explanation signals capture task-relevant temporal, spectral, and spatial patterns compared to the raw EEG. The figure summarizes results across methods, highlighting that Pattern-Local produces more distinct cue-locked responses and source activations consistent with known motor-imagery physiology.

# F  Additional Discussion

## F.1  Evaluation metrics

Many studies in explainable AI evaluate attribution methods using faithfulness-based metrics such as insertion, deletion, or accuracy drop. These metrics measure how the model's output changes when parts of the input are removed or revealed. They do not use ground-truth information about which features are relevant, but instead reward explanations that match the model's predictive behavior. On suppressor-aware benchmarks, this is problematic. Removing these features still alters the model's output if a model relies on suppressor variables due to statistical dependencies rather than the true signal. As a result, faithfulness metrics can incorrectly reward attributions that highlight suppressors. This behavior follows from how such metrics are defined and is not simply an implementation artifact.

Wilming et al. [35] provide a theoretical analysis of suppressor variables and show that many popular XAI methods assign non-zero importance to suppressor features when the data contain correlated noise. Blücher et al. [6] further show that the PredDiff method, which is based on conditional expectations, is closely connected to Shapley values. Since Shapley values distribute credit according to statistical dependence rather than causal relevance, they can also assign importance to suppressors or other correlated but task-irrelevant features. Therefore, evaluation metrics implicitly following this logic cannot distinguish between true causal relevance and statistical influence. They may judge an explanation as "faithful" even when it highlights misleading parts of the input.

Haufe et al. [13] argue that faithfulness to the model is insufficient for a correct explanation and that relying solely on it can mislead scientific interpretation or model validation. They call for formal criteria of correctness that go beyond reproducing model behavior. It is concerning that many of the most commonly used XAI methods and evaluation metrics share the same weakness: both can appear consistent with the model while failing to reflect the underlying ground truth.

We also include the mean squared error (MSE) in our ablation studies for completeness. MSE does not capture spatial structure and penalizes false positives and negatives equally, yet it produces the same ranking of methods as our primary metrics. This consistency supports the robustness of our findings. Overall, metrics that do not use ground-truth relevance, such as faithfulness-based scores, can reward faithful but misleading explanations. In suppressor-aware settings, this risk is particularly severe. Ground-truth-based metrics are therefore more appropriate for evaluating whether an explanation truly identifies the relevant features.

# G  Ablation Studies

## G.1  Mixing white and correlated noise

Instead of varying the signal-to-noise ratio (SNR), we can fix the SNR and introduce a new parameter $\beta$ to control the balance between white and correlated noise. Specifically, $\beta = 0$ corresponds to the purely white-noise scenario (WHITE), while $\beta = 1$ corresponds to the purely correlated-noise scenario (CORR). Under this setup, the data-generation process can be written as

$$\mathbf{x}^{(n)} = \alpha\big(R^{(n)} \circ (H \circ \mathbf{a}^{(n)})\big) + (1 - \alpha)\big(\beta(G \circ \boldsymbol{\eta}_1^{(n)}) + (1 - \beta)\boldsymbol{\eta}_2^{(n)}\big), \tag{15}$$

where $\alpha$ controls the SNR, $\beta$ governs the relative contributions of white versus correlated noise, $G$ is the same Gaussian spatial filter used in the CORR scenario, and $\boldsymbol{\eta}_1^{(n)}, \boldsymbol{\eta}_2^{(n)}$ represent the background noise sources.

Figure 13 shows the XOR scenario under this new scheme, demonstrating that as $\beta$ increases (i.e., as correlated noise becomes more dominant), the performance gap between LIME and PatternLocal grows accordingly.

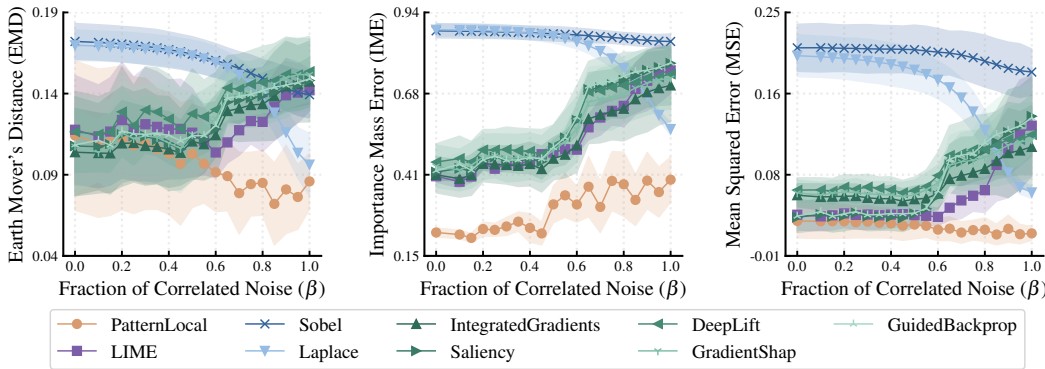

Figure 13: Comparison of methods in the new XOR scenario at a fixed SNR of $\alpha = 0.2$, with $\beta$ varying from 0 (WHITE) to 1 (CORR). Results are shown for the three metrics: Earth Mover's Distance (EMD), Importance Mass Error (IME), and Mean Squared Error (MSE). As $\beta$ increases, the performance of all methods decreases, except for PatternLocal, which remains stable.

## G.2  Model architecture and hyperparameter objective

Figures 14–17 summarise the ablation study carried out in this appendix. For every *explanation method* under investigation we systematically vary two experimental factors:

- **Model architecture.** We compare a multi-layer perceptron (**MLP**) with a convolutional neural network (**CNN**).
- **Hyper-parameter objective.** Method hyperparameters are tuned either for minimum Earth Mover's Distance (**EMD**) or for minimum mean squared error (**MSE**) on the validation set.

This yields the four result groups

- MLP + EMD (Figure 14)
- MLP + MSE (Figure 15)
- CNN + EMD (Figure 16)
- CNN + MSE (Figure 17)

Each figure contains three panels that report, as a function of the signal-to-noise ratio (SNR), the Earth Mover's Distance (top), the Importance Mass Error (middle), and, included here for completeness, the Mean Squared Error (bottom). Shaded bands correspond to one standard deviation across the test images.

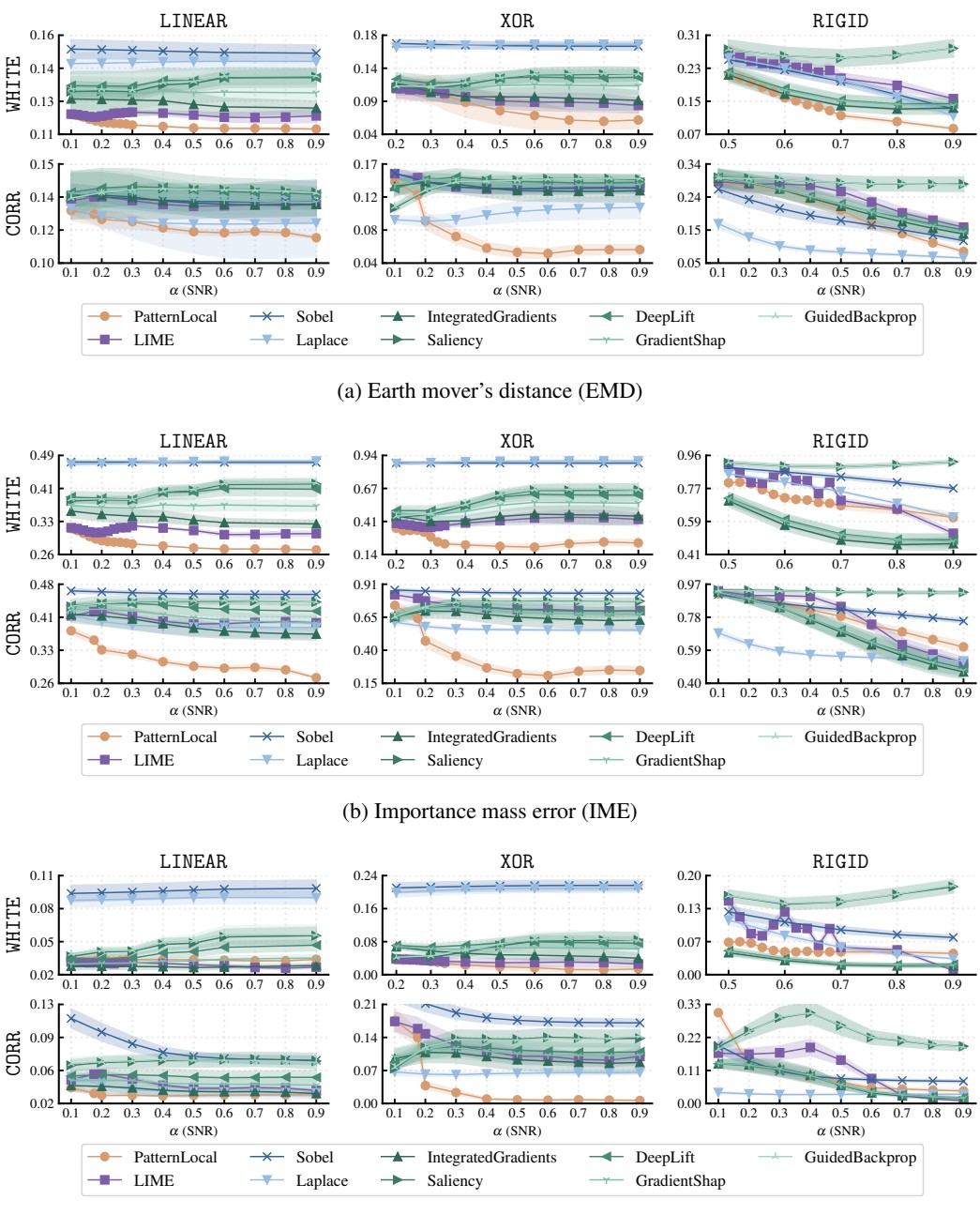

(a) Earth mover's distance (EMD)

(b) Importance mass error (IME)

(c) Mean squared error (MSE)

Figure 14: Quantitative evaluation of feature-importance maps for the **MLP model** on $8 \times 8$ XAI-TRIS benchmark images generated by several methods as a function of the SNR. The classifier attains at least $90\%$ accuracy and all method hyperparameters were **tuned for EMD**. We take $\mathbf{h}_{\mathbf{x}_\star}(\cdot) = \mathrm{id}$ and $R(\cdot) = Q(\cdot) = \|\cdot\|_2^2$. Panels (a)–(c) report EMD, IME, and MSE, respectively; shaded bands denote one standard deviation.

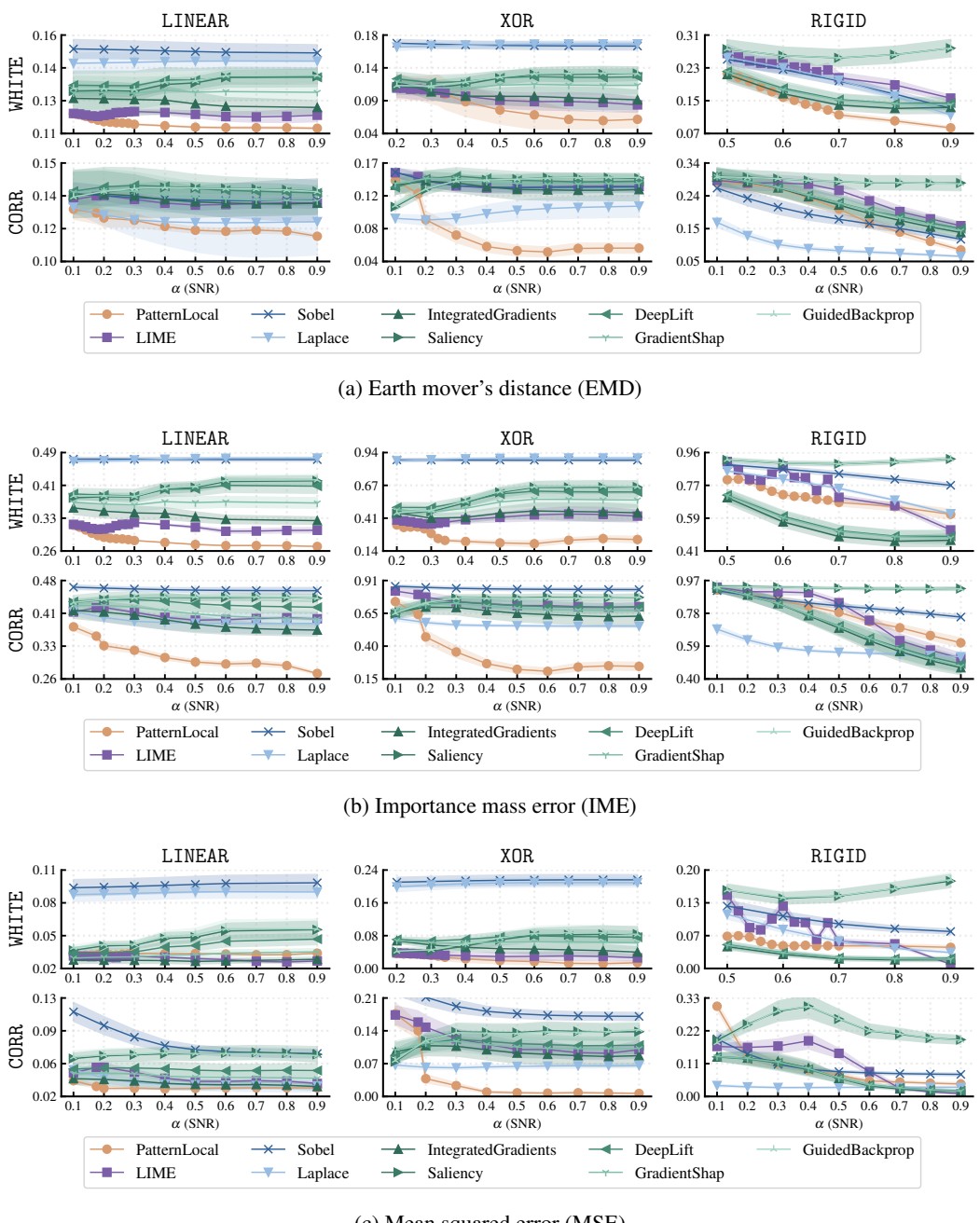

(a) Earth mover's distance (EMD)

(b) Importance mass error (IME)

(c) Mean squared error (MSE)

Figure 15: Quantitative evaluation of feature-importance maps for the **MLP model** on $8 \times 8$ XAI-TRIS benchmark images, with method hyperparameters **tuned for MSE**. All remaining settings match Fig. 14.

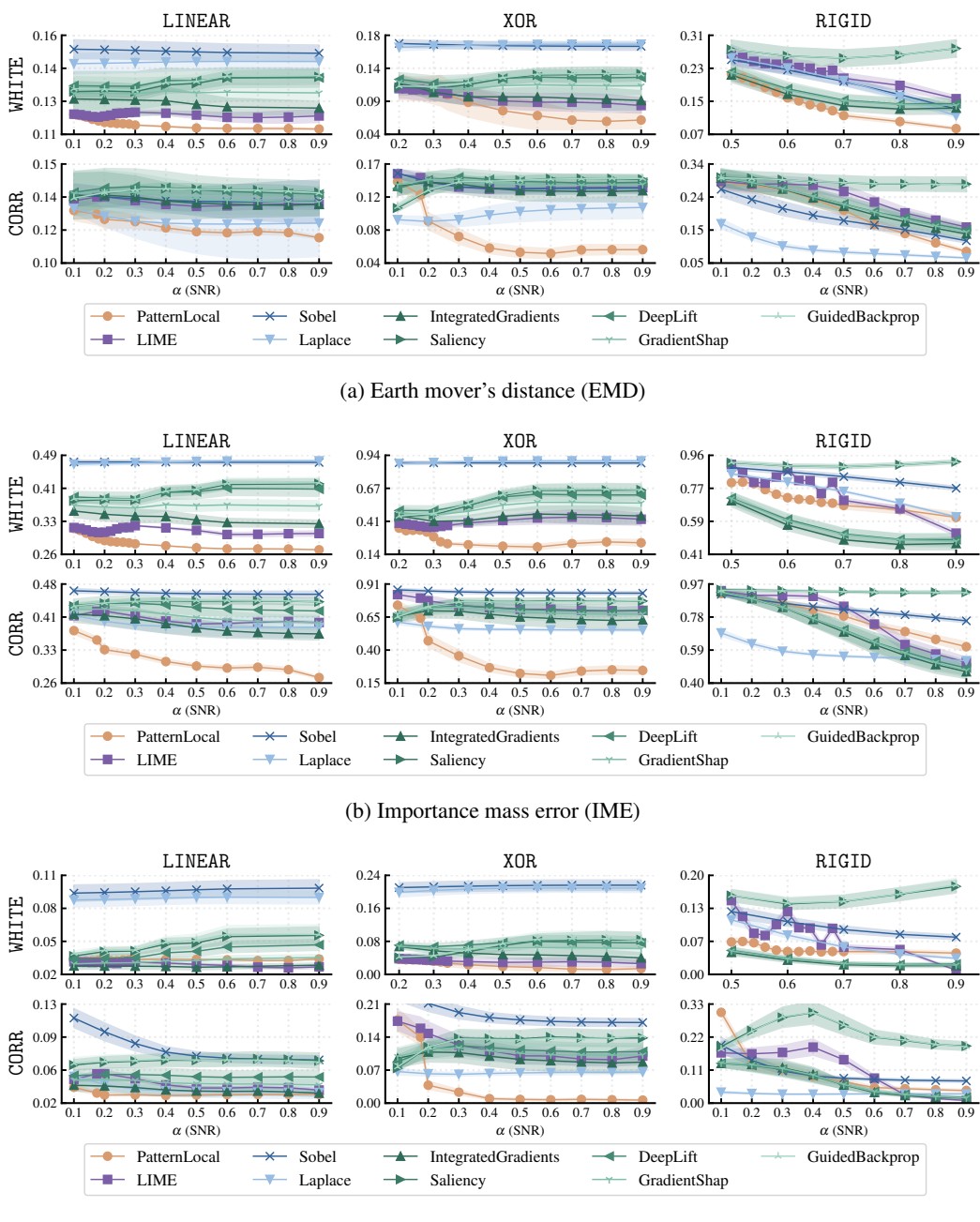

(a) Earth mover's distance (EMD)

(b) Importance mass error (IME)

(c) Mean squared error (MSE)

Figure 16: Quantitative evaluation of feature-importance maps for the **CNN model** on $8 \times 8$ XAI-TRIS benchmark images with hyperparameters **tuned for EMD**. The network achieves at least $90\%$ accuracy; other settings as in Fig. 14.

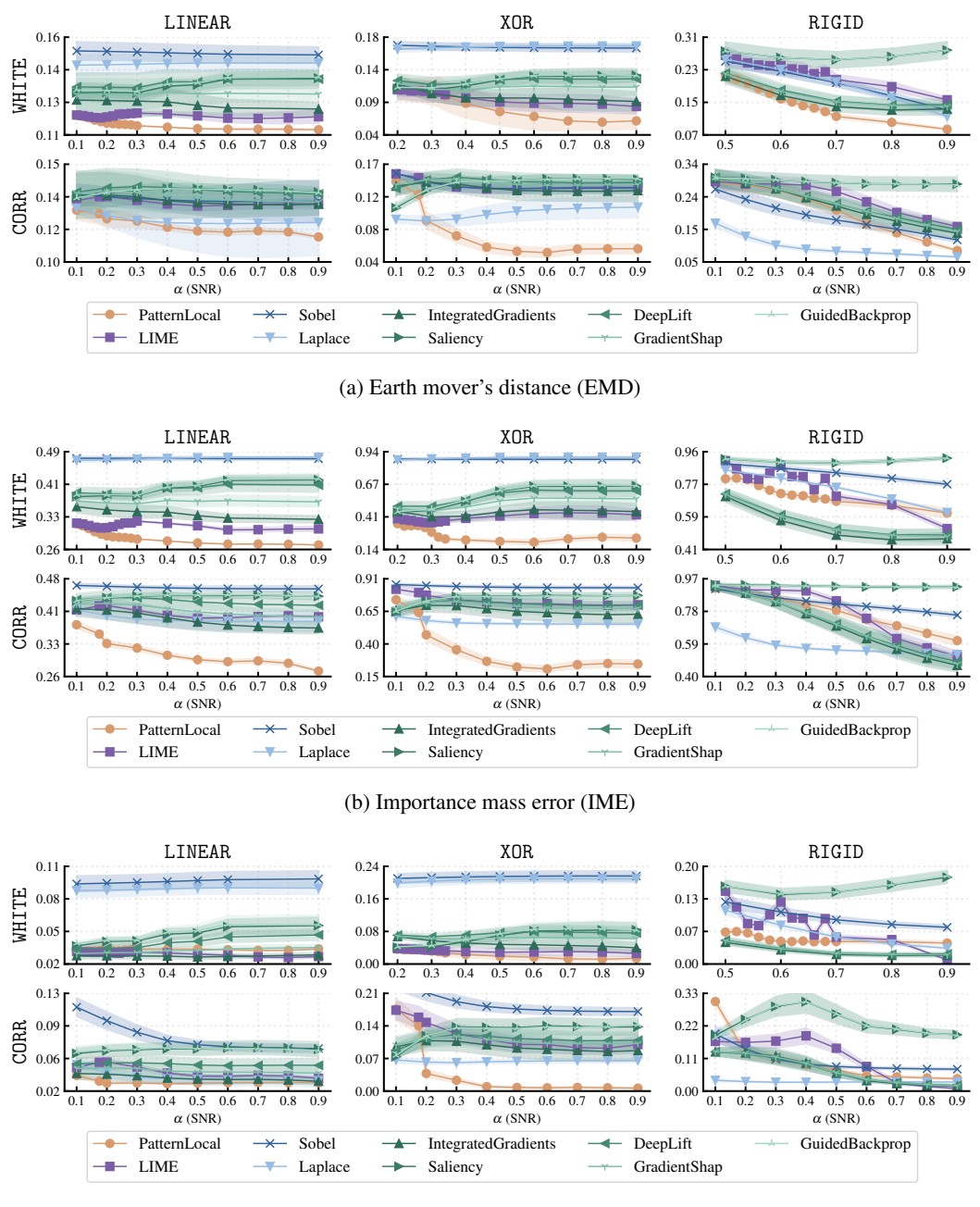

(a) Earth mover's distance (EMD)

(b) Importance mass error (IME)

(c) Mean squared error (MSE)

Figure 17: Quantitative evaluation of feature-importance maps for the **CNN model** with hyperparameters **tuned for MSE**. Experimental conditions mirror those of Fig. 14.

