# OpenReview forum: "Minimizing False-Positive Attributions in Explanations of Non-Linear Models"
_NeurIPS.cc/2025/Conference — NeurIPS 2025 poster_

### Official Review · Reviewer_7PSJ · 2025-06-10

**Clarity:** 2
**Significance:** 3
**Originality:** 3
**Rating:** 4
**Confidence:** 4

**Summary:**

The manuscript introduces a new XAI method for vision tasks called PatternLocal. Similar to previous methods such as LIME, the proposed PatternLocal method constructs a local linear surrogate, and then constructs a generative model using the resulting weights. It is claimed that the process mitigates the effects of suppressor variables in generating explanations (false-positive attribution).

**Questions:**

> Please explain the expected behavior of each plot as a function of the x-axis variable (alpha).
> Please see the weaknesses section for additional comments on the manuscript and organization, and provide clarifications in your rebuttal if needed.

**Ethical Concerns:**

["NO or VERY MINOR ethics concerns only"]

**Final Justification:**

As mentioned in my response to the author's rebuttal, the rebuttal has addressed some of my concerns and technical questions.

The authors have also mentioned that they will address the notational issues in the original manuscript prior to final publication, and their described plan of action is satisfactory to address many of the issues. However, it would have been more helpful if these issues were addressed prior to the original submission as the reviewers will not have the opportunity to review the modified manuscript afterwards.

Given the above, I will increase my score to Borderline Accept.

**Quality:**

3

**Strengths And Weaknesses:**

Strengths:
>- The topic is timely and the issue of false-positive attribution (particularly due to suppressor variables) is well known and solutions such as those provided in this work are of interest.
>- The examples, such as the Toy Example of Section 3.2 and the example in Section 2.2 of the suppressor variable are helpful and well-explained.

Weaknesses: The manuscript is generally not well-written. The notation is not consistent and the explanations are incomplete and vague. More detailed comments are provided below:
>- On Page 3, Line 67, $h_{x_*}$ is introduced as a mapping that produces a low-dimensional semantically meaningful representation. The definition is vague and needs additional explanation. Furthermore, the output is denoted as $x'_*$, however, in Equation 2, the input to the inverse $h^{-1}$ is shown by z (which seems more suitable). The notational inconsistency is repeated in many other instances and reduces the readability of the manuscript.
>- It is not clear why on Page 3, Line 94, w is used instead of W, whereas in all previous instances, W was used to represent the transform matrix.
>- In Page 3, Equation 2, $\hat{s}$ is recovered with respect to which loss metric? (e.g., is it to minimize the mean squared error?)
>- The manuscript is not well-organized. For instance, the advantages of PatternLocal are discussed in Page 4 in detail before the method is introduced even at a high level. It seems more appropriate to provide a high level description of the main ideas in PatterLocal first, and then describe its advantages compared to existing methods.
>- In Figure 4, it is not clear what the x-axis represents. It is stated in the caption that it represents the SNR, however, the value ranges from 0 to 1. It seems from the supplementary material that it represents alpha (as a surrogate for SNR). If so, should we expect that the plots be decreasing in alpha? (as the noise decreases) If so, why are some of the plots increasing in alpha?

---

> ### Author Rebuttal · Authors · 2025-07-31
>
> # Reviewer 7PSJ
>
> Thank you for emphasizing the timeliness of addressing false-positive attributions and for appreciating our explanatory examples. As you and others noted, notation inconsistencies, figure clarity, and manuscript organization need improvement. **Note, some subscripts were omitted to avoid OpenReview formatting issues.**
>
> ## Questions and Comments
>
> > "On Page 3, Line 67, $h$ is introduced as a mapping that produces a low-dimensional semantically meaningful representation. The definition is vague and needs additional explanation. Furthermore, the output is denoted as $x_\star'$ , however, in Equation 2, the input to the inverse $h^{-1}$ is shown by z (which seems more suitable). The notational inconsistency is repeated in many other instances and reduces the readability of the manuscript."
> >
>
> We thank Reviewer 7PSJ for pointing out the notational issues. Below, we clarify the mapping $h$, its role in Eq. 5 and Eq. 6, and the changes we've made to fix the inconsistencies.
>
> The map $h: \mathcal X \to \mathbb R^{D'}$ (changed to $h: \mathbb R^{D} \to \mathbb R^{D'}$) defines an instance-specific simplification: for a given input $\mathbf{x}_\star$, it maps any other input $\mathbf{x}$ (real) or $\mathbf{z}$ (synthetic) to a lower-dimensional, interpretable space via $x' = h(\mathbf{x})$. The form of $h$ depends on the domain (e.g., superpixels or low-rank projection), and we keep the subscript to show it's tied to the reference instance.
>
> To simplify notation, we now use a prime $'$ to mark variables in the reduced space e.g., $\mathbf{x}'$, $\mathbf{z}'$, $D'$. Real data from $\mathcal D$ are denoted $\mathbf{x}$, and synthetic samples are $\mathbf{z}$, giving:
>
> $\mathbf{x}' = h(\mathbf{x}), \quad \mathbf{z}' = h(\mathbf{z})$
>
> This holds for Eq. 5 (LIME/KernelSHAP on synthetics) and Eq. 6 (PatternLocal on real data). We also define the inverse consistently:
>
> $h^{-1}: \mathbb{R}^{D'} \to \mathbb{R}^{D}, \quad h: \mathbb{R}^{D} \to \mathbb{R}^{D'}$
>
> For LIME, $h^{-1}(\mathbf{z}')$ fills zeros with a reference value $r$ and upsamples the mask back to $\mathbb R^{D}$, exactly as in Ribeiro et al. (2016). In the low-rank case, it's the linear decoder. PatternLocal does not use synethetic samples but uses $h$ to map real samples to simplified sapce before optimization.
>
> We've added a one-page notation table in the appendix (linked from Section 2.1) to aid readability. We hope these updates address the reviewer’s concerns.
>
> | Symbol | Description |
> | --- | --- |
> | $\mathcal{X} \subset \mathbb{R}^D$ | Input space of dimension $D$. |
> | $\mathcal{Y} \subset \mathbb{R}$ | Output space (scalar). |
> | $\mathbf{x} \in \mathcal{X},\; y \in \mathcal{Y}$ | A data point and its label. |
> | $\mathbf{x}^{\star}$ | The specific instance whose prediction $f(\mathbf{x}^{\star})$ we wish to explain. |
> | $D,\; D'$ | $D$: input dimensionality; $D' \ll D$: simplified input dimensionality. |
> | $\mathbf{x}' = \mathbf{h}(\mathbf{x})$ | Simplified/semantic representation of data point $\mathbf{x}$. |
> | $\mathbf{h} : \mathbb{R}^{D} \to \mathbb{R}^{D'}$, $\mathbf{h}^{-1} : \mathbb{R}^{D'} \to \mathbb{R}^{D}$ | Forward and inverse mapping between original and simplified spaces. |
> | $f$ | The original model being explained, $f : \mathcal{X} \to \mathcal{Y}$. |
> | $g$ | Interpretable surrogate model fit locally around $x^{\star}$. |
> | $\mathcal{D} \subset \mathcal{X}$ | Training data used by PatternLocal for local statistics. |
> | $\Pi_{x'_{\star}}(\cdot)$ | Locality kernel down-weighting samples distant from $\mathbf{x}'_{\star}$. |
> | $R(\cdot)$, $Q(\cdot)$ | Regularization in LIME and PatternLocal. |
> | $\lambda$ | Regularization strength. |
> | $\alpha$ | Signal-to-noise mixing coefficient for synthetic data. |
> | $\mathbf{z}'$, $\mathcal{Z}$ | Simplified artificial samples and their space. |
> | $\mathbf{s}$, $\mathbf{s}^{\text{LIME}}$, $\mathbf{s}^{\text{PatternLocal}}$ | Feature-importance scores (saliency maps); either generic or for specific method. |
> | $\mathbf{w}$ | Coefficient vector from linear surrogate models (Eq. 5). |
> | $\mathbf{a}$ | PatternLocal's explanation vector via forward-model regression (Eq. 6). |
> | $\tilde{y} = \mathbf{w}^{\top} \mathbf{h}_{\mathbf{x}^{\star}}(\mathbf{x})$ | Surrogate prediction in PatternLocal. |
> | $\mathbf{u}$, $\mathbf{v}$ | Regression coefficients that minimize Eq. 5 or Eq. 6 loss (equals $\mathbf{a}$ or $\mathbf{w}$ at optimum). |
> | $\mathbf{W}$ | Backward weight matrix: $\hat{\mathbf{m}} = \mathbf{W}^{\top} \mathbf{x}$. |
> | $\Sigma_X$ | Covariance matrix of input $\mathbf{x}$, used in $A = \Sigma_X W \Sigma_M^{-1}$. |
> | $x_1, x_2, x_3$ | Features in the XOR example; $x_3$ is a suppressor variable. |
> | $\mathcal{F}^+(x_\star)$ | Ground-truth relevant pixels for data point. |
>
> ---
>
> > "It is not clear why on Page 3, Line 94, w is used instead of W, whereas in all previous instances, W was used to represent the transform matrix."
> >
>
> Thank you. It has been changed to $\mathbf{W}$.
>
> ---
>
> > ""In Page 3, Equation 2, $\hat{s}$ is recovered with respect to which loss metric? (e.g., is it to minimize the mean squared error?)
> >
>
> Eq. 2 states a linear projection; the meaning of the recovered $\hat{\mathbf{s}}$ (renamed to $\hat{\mathbf{m}}$) and the implicit "loss metric" come entirely from the way $\mathbf{W}$ was learned. It could be recovered with minimizing squared error like in regression or by maximizing class separation like LDA. A deeper explanation is provided in (Haufe et. al. 2014).
>
> ---
>
> > "The manuscript is not well-organized. For instance, the advantages of PatternLocal are discussed in Page 4 in detail before the method is introduced even at a high level..."
> >
>
> We apologize for the extra time reviewing this may have caused. We have included a high-level algorithmic explanation of the method (as requested by Reviewer zrTT) beforehand. The order in Section 3 is thus now. High-level description → Advantages → Formal Objective → Toy Example.
>
> | Step | Description |
> | --- | --- |
> | **1. Fit local discriminative surrogate** | Pick a linear explainer (e.g., LIME, KernelSHAP, first-order gradient) and solve its optimization to obtain the weight vector $\mathbf{w}$ that best approximates the model's decision boundary around the point $\mathbf{x}_\star$ (Eq. 5). |
> | **2. Construct data neighborhood** | From the training set, collect samples near $\mathbf{x}_\star$ and weight them using the same locality kernel $\Pi$. |
> | **3. Transform surrogate into generative pattern** | Regress the surrogate scores $\hat{y}=\mathbf{w}^\top h (\mathbf{x})$ back onto the simplified inputs $h (\mathbf{x})$  using the weighted neighborhood. The resulting coefficient vector $\mathbf{a}$ (Eq. 6) defines a generative pattern that minimizes suppressor variables. |
> | **4. Produce saliency / importance map** | Return $\mathbf{a}$ as the PatternLocal explanation for $\mathbf{x}_\star$. If a simplified representation (e.g. super pixels) was used, expand $\mathbf{a}$ back to the input resolution. |
>
> Other changes include a dedicated Related Works section, notational fixes, further clarification on notation and more than can be seen in the response to the other reviewers.
>
> ---
>
> > "In Figure 4, it is not clear what the x-axis represents. It is stated in the caption that it represents the SNR, however, the value ranges from 0 to 1. It seems from the supplementary material that it represents alpha (as a surrogate for SNR). If so, should we expect that the plots be decreasing in alpha? (as the noise decreases) If so, why are some of the plots increasing in alpha? … Please explain the expected behavior of each plot as a function of the x-axis variable (alpha)."
> >
>
> Thanks for pointing out the missing axis label. We've added $\alpha$ to the x-axis in Figures 4, 7, 12, 13, and 15. Signal-to-noise ratio (SNR) is controlled by $\alpha \in [0, 1]$: higher $\alpha$ means stronger signal. Not all curves decrease with $\alpha$ due to two effects:
>
> 1. **Edge-detector baselines (Sobel, Laplace):**
>    As $\alpha$ increases, tetromino edges sharpen, and filters respond more outside true boundaries. This leads to more false positives, raising EMD/IME.
>
> 2. **LIME, PatternLocal, etc.:**
>    Their scores generally improve with $\alpha$, since higher SNR aids local fit. The spike at $\alpha \approx 0.9$ (WHITE-LINEAR) is likely a numerical issue—at high $\alpha$, the dataset becomes rank-deficient, making LIME's regression (and thus PatternLocal) unstable.
>
> Empirical rank of selected datasets:
>
> | $\alpha$ | 0.1 | 0.2 | 0.3 | 0.4 | 0.5 | 0.6 | 0.7 | 0.8 | 0.9 |
> | --- | --- | --- | --- | --- | --- | --- | --- | --- | --- |
> | **XAI-TRIS WHITE LINEAR** | 4096 | 4096 | 4096 | 4096 | 4096 | 4096 | 4096 | 3554 | **2** |
>
> | $\alpha$ | 0.5 | 0.6 | 0.7 | 0.8 | 0.9 |
> | --- | --- | --- | --- | --- | --- |
> | **XAI-TRIS WHITE RIGID** | 4096 | 4096 | 4096 | 4096 | 2881 |
>
> The drop to rank 2 explains the WHITE-LINEAR spike. RIGID stays full-rank longer, so its curve stays smooth. We’ve added this to the manuscript and now warn that high $\alpha$ may cause rank-deficiency.
>
> ---
>
> ## Final note
>
> We thank Reviewer 7PSJ once more for their constructive feedback. As detailed above, we have answered every question, clarified all ambiguities and updated the manuscript accordingly. We therefore kindly ask you to reconsider your scores in light of these improvements. Should any further points arise, we are happy to provide additional clarifications at any time.

---

> > ### Comment · Reviewer_7PSJ · 2025-07-31
> >
> > I thank the authors for the comprehensive response.
> >
> > The rebuttal has addressed some of my concerns. Specifically, the response regarding Figure 4 and direction of change with respect to alpha is insightful.
> >
> > The authors have mentioned that they will address the notational issues in the original manuscript prior to final publication, and their described plan of action is satisfactory to address many of the issues. However, it would have been more helpful if these issues were addressed prior to the original submission as the reviewers will not have the opportunity to review the modified manuscript afterwards.
> >
> > Given the above, I will increase my score to Borderline Accept.

---

### Official Review · Reviewer_Xj2D · 2025-06-30

**Clarity:** 3
**Significance:** 3
**Originality:** 3
**Rating:** 5
**Confidence:** 4

**Summary:**

This study tackles the problem that existing XAI methods tend to assign high importance to suppressor variables—features that are not directly related to the prediction target but cancel out noise and thereby boost the model’s overall accuracy.
Previous work has proposed a method to suppress the influence of suppressor variables in linear models.
The proposed method in this study extends those ideas so they can be applied to non-linear predictive models.
Concretely, around the test instance to be explained, the method trains a local linear surrogate model of the non-linear predictor using LIME.
It then estimates parameters that map the surrogate model’s outputs back to simplified inputs of that test instance, and treats those estimated parameters as the feature importances.
Experiments on datasets constructed to contain suppressor variables demonstrate that the proposed approach produces correct explanations, outperforming existing feature attribution methods.

**Questions:**

If you have rebuttals to the points listed under Weaknesses, please present them.

**Ethical Concerns:**

["NO or VERY MINOR ethics concerns only"]

**Final Justification:**

The authors' explanations for the points I raised were convincing, and my concerns have been resolved. Therefore, I raised my score.

**Limitations:**

Yes.

**Paper Formatting Concerns:**

None.

**Quality:**

3

**Strengths And Weaknesses:**

# Strengths
- Extending the Pattern proposed by Haufe et al. to work with non-linear models is both novel and practically valuable, and the derivation of the proposed method appears sound.
- The toy example convincingly shows that—even for a non-linear model—the proposed approach can drive the importance of a suppressor variable to zero, effectively illustrating the method’s validity.
- On the XAI-TRIS benchmark, the proposed method performs better than many alternative explanation techniques on images corrupted by noise and suppressor variables, demonstrating its usefulness.

# Weaknesses
- It is unclear why simply adding an L1 regularizer in LIME would not suffice to push suppressor-variable importance toward zero. In the experiments, both LIME and the proposed method seem to use L2 regularization, yet the behavior under L1 regularization is not examined in depth.
- The evaluation metrics employed are not standard. For image classification, metrics such as Insertion, Deletion, or Infidelity are commonly used to measure explanatory faithfulness, whereas the paper adopts a non-standard metric called EMD; no compelling reason is given for this choice.
-The validity of that metric is questionable. In (10), defining EMD, |\mathbf{S}| appears to denote the absolute values of the estimated importance map, but taking absolute values discards whether a feature contributes positively or negatively. In IME (11), the numerator and denominator are summed independently, so it is unclear how false positives and false negatives are captured. Moreover, the meaning of superscripts such as |\mathcal{F}^+| and |\mathbf{s}| is ambiguous—do they denote set cardinality, absolute value, or vector norm?

---

> ### Author Rebuttal · Authors · 2025-07-31
>
> # Reviewer Xj2D
>
> Thank you for recognizing the novelty and practical value of extending the Pattern method to non-linear settings and for noting the clarity of our toy example. We share your interest in L1-regularized baselines and standard faithfulness metrics. We therefore begin by clarifying notation, elaborating on metric choice, and adding new experiments. **Note, some subscripts were omitted to avoid OpenReview formatting issues.**
>
> ## Questions and Comments
>
> > "It is unclear why simply adding an L1 regularizer in LIME would not suffice to push suppressor-variable importance toward zero. In the experiments, both LIME and the proposed method seem to use L2 regularization, yet the behavior under L1 regularization is not examined in depth."
> >
>
> An $\ell_1$ penalty encourages sparsity but does not distinguish between suppressors and informative features. In the toy XOR example and on XAI-TRIS, LIME with $\ell_1$ regularization reduces the magnitude of all coefficients but continues to assign non-zero importance to suppressors. PatternLocal, in contrast, transforms the discriminative surrogate into a generative pattern that down-weights features uncorrelated with the surrogate output. In the revised paper, we include in Appendix D: Extended results experiments comparing LIME with $\ell_1$ regularization to PatternLocal. Across all tasks PatternLocal drives suppressor attributions closer to zero and have lower EMD and IME scores. This indicate that PatternLocal adds value beyond sparsity.
>
> ---
>
> > "The evaluation metrics employed are not standard. For image classification, metrics such as Insertion, Deletion, or Infidelity are commonly used to measure explanatory faithfulness, whereas the paper adopts a non-standard metric called EMD; no compelling reason is given for this choice. The validity of that metric is questionable."
> >
>
> We chose EMD and IME because they are tailored to suppressor-aware benchmarks: EMD measures the spatial cost of moving attribution mass to ground-truth locations, and IME penalizes false positives more than false negatives. These metrics were introduced by (Clark et al. 2024) for the XAI-TRIS benchmark datasets. In Appendix E: Ablation Studies, we have already included the MSE measure, which aligns with the conclusion drawn from the EMD and IME metric. Nonetheless, we agree and justify this choice more in Section 4.4. We have added metrics on local fidelity of the surrogate model as the weighted error as requested by Reviewer zNrP
>
> Regarding faithfulness, metrics like insertion and deletion, on the other hand, are not based on an explicit ground-truth. However, they implicitly incentivize attribution to suppressors, which we criticize. In fact, faithfulness metrics are strongly linked and, under some conditions, equivalent to Shapley values for attribution, as shown by (Wilming et al, 2023, Theoretical Behavior of XAI Methods in the Presence of Suppressor Variables) and (Blücher et al, 2022, PredDiff: Explanations and interactions from conditional expectations).
>
> In short, we do not believe that faithfulness metrics should be considered the gold standard to evaluate attributions as faithful attributions can lead to incorrect interpretations and mistakes in downstream tasks such as scientific discovery or model validation as also noted in (Haufe et al 2024, Explainable AI needs formal notions of explanation correctness). Therefore, we see it as a problematic development in the field that both the most prominent XAI methods and the most common metrics to evaluate them share the same limitations.
>
> ---
>
> > "In (10), defining EMD, $|s|$ appears to denote the absolute values of the estimated importance map, but taking absolute values discards whether a feature contributes positively or negatively. In IME (11), the numerator and denominator are summed independently, so it is unclear how false positives and false negatives are captured"
> >
>
> The absolute‐value notation in Eq. 10 and the independent sums in Eq. 11 are intentional. For the XAI-TRIS benchmark we only care whether a feature is highlighted at all, not whether its contribution is numerically positive or negative. In addition, in many cases (XAI-TRIS LINEAR and XAI-TRIS XOR), the absence of one feature is as informative as the presence of another feature. By not taking the absolute value, we would penalize models for this.So in short, measuring unsigned mass therefore lets Earth Mover’s Distance (EMD) and Importance Mass Error (IME) focus on the presence or absence of attribution while avoiding unnecessary penalties.
>
>
> > "Moreover, the meaning of superscripts such as $|\mathcal{F}^+|$ and $|\mathbf{s}|$ is ambiguous—do they denote set cardinality, absolute value, or vector norm?"
> >
>
> Thank you for highlighting this ambiguity. It was also noted by Reviewer zrTT. We use $|\mathbf{s}|$ to refer the absolute value in Eq. 10 and within the summation in Eq. 11. We instead use the notation $\text{card}(\cdot)$ to refer to the cardinality e.g. $\text{card}(\mathcal{F}^+)$ and $\text{card}(\mathbf{s})$.
>
> Furthermore, we have defined $\mathcal{F}^+$ in Appendix B, Eq. 13 and 14. but have now included the definition in Section 4.4. It is defined as the set of ground-truth salient pixels for the model's prediction, following (Clark et al, 2024).
>
> ---
>
> ## Final note
>
> We thank Reviewer Xj2D once more for their constructive feedback. As detailed above, we have answered every question, clarified all ambiguities and updated the manuscript accordingly. We therefore kindly ask you to reconsider your scores in light of these improvements. Should any further points arise, we are happy to provide additional clarifications at any time.

---

> > ### Comment · Reviewer_Xj2D · 2025-08-06
> >
> > Thank you for the response. Your explanations for the points I raised were convincing, and my concerns have been resolved. I intend to raise my score.

---

### Official Review · Reviewer_zNrp · 2025-07-01

**Clarity:** 3
**Significance:** 2
**Originality:** 2
**Rating:** 4
**Confidence:** 4

**Summary:**

The paper studies the phenomenon of  “suppressor variables”, which are features that influence model predictions but are not statistically dependent on the output variable, potentially leading to false-positive feature attributions.

The authors propose PatternLocal, a novel  and model agnostic XAI technique that generalizes the Pattern approach into the local, non-linear setting. PatternLocal operates in three steps: it (1) builds a locally linear surrogate model around an instance of interest (via LIME, SHAP, or gradients), (2) transforms the surrogate’s discriminative weights into a generative explanation using local data statistics, and (3) outputs instance-level attributions that suppress the effect of suppressor variables while preserving local model fidelity.

The authors conduct empirical validation on the XAI-TRIS benchmark and an artificial lesion MRI dataset, which are specialized benchmarks that offer suppressor-variable scenarios. After hyperparemter tuning, PatternLocal tends to outperforms established XAI methods on the Earth Mover’s Distance (EMD) and Importance Mass Error (IME) metric.

**Questions:**

Why focus on Earth Mover’s Distance (EMD) and Importance Mass Error (IME) and not simply using more standard metrics like MSE or PSNR for  visual explanation comparison? How sensitive are your conclusions to metric selection?

The paper uses exhaustive hyperparameter tuning for all methods. Is this necessary for PatternLocal to perform well, or do optimal hyperparameters at least generalize in some way? How does PatternLocal perform in more typical, default or lightly-tuned scenarios compared to common default settings of existing explanations?

What kind of explanations does PatternLocal produce when be applied to datasets like ImageNet or tabular data, where finding sufficient, well-aligned local neighbors is difficult?

 Could access to generative models or learned latent spaces provide a substitute for local neighborhood samples where data is limited or not aligned? Would such extensions preserve PatternLocal’s suppressor-mitigation properties?

**Ethical Concerns:**

["NO or VERY MINOR ethics concerns only"]

**Final Justification:**

My initial score was 3 and I have increased it to 4 after reading the authors response. However, I still have some doubt regarding the overall utiliy of the method as requires standardized views and the existance of sufficiently similar samples in the local neighborhood.  Providing more evidence on how the method can be applied more broadly would substantially strengthen the paper.

**Limitations:**

yes

**Paper Formatting Concerns:**

No major formatting issues.

**Quality:**

3

**Strengths And Weaknesses:**

Strengths:

-	Studying the behavior of explanation methods in the presence of “suppressor variables” is relevant.

-	The paper proposes a theoretical sound way to extend Pattern approach to non-linear models by building on top of existing (model agnostic) explanation approaches.

-	The paper is well written and easy to follow.

Weaknesses:

- The theoretical contributions are limited as the method is a direct extension of the existing Pattern method.

-	The method appears to be computationally way more expensive compared to the baselines and the experimental results only partly justify the effort despite extensive hyperparameter search.

- All experiments are primarily on synthetic, or semi-synthetic image data. The generalization to complex real-world datasets remains unclear.

---

> ### Author Rebuttal · Authors · 2025-07-31
>
> # Reviewer zNrP
>
> Thank you for recognizing the relevance of suppressor-variable mitigation, the soundness of our derivation, and the careful experimental design. We address each of your concerns below and describe the changes made to the paper.  **Note, some subscripts were omitted to avoid OpenReview formatting issues.**
>
> ## Questions and Comments
>
> ---
>
> > "The theoretical contributions are limited as the method is a direct extension of the existing Pattern method."
> >
>
> While PatternLocal builds on the insight of transforming discriminative weights into generative patterns, extending this idea to local surrogates requires a non-trivial adaptation. The revised paper now explicitly derives the kernel-weighted regression objective (Eq. 6) and its closed-form solution for $\ell_2$ regularization (Eq. 7). We further provide a discussion on sample complexity (as requested by Reviewer zrTT) and the conditions under which the local regression yields unbiased estimates. These theoretical additions clarify how PatternLocal generalizes Pattern to non-linear settings and highlight its novelty.
>
> It is important to note that an explicit matrix covariance estimate (and its multiplication) is not always required, especially when the surrogate already contains an inverse-covariance form. The core step in PatternLocal is the kernel-weighted moment
>
> $a_{\ell2} = \frac{\text{Cov}\bigl[h(\mathbf{x}), \hat{y}\bigr]}{\text{Var}[\hat{y}] + \lambda}$ where $\hat{y} = \mathbf{w}^\top h(\mathbf{x})$
>
> which for binary classification only needs (i) the scalar local variance of $\hat y$ and (ii) the vector cross-covariance with the simplified features $h_{\mathbf{x}_\star}(\mathbf{x})$. We estimate those moments with a Nadaraya–Watson (NW) kernel smoother. For any function $g$
>
> $\hat{\mathbb{E}}[g] = \frac{\sum_{i=1}^{n^{\text{loc}}} K(\|h(\mathbf{x}(i))-\mathbf{0}\|) g(\mathbf{x}(i))}{\sum_{i=1}^{n^{\text{loc}}} K (\|h(\mathbf{x}(i))-\mathbf{0}\|)}$,
>
> where $K_\sigma$ is an isotropic kernel of bandwidth $\sigma$ in the $D'$-dimensional interpretable space and $n_{\text{loc}}$ the number of samples in the local neighborhood.  Nadaraya–Watson analysis (Yin et al, 2010) gives that the MSE is minimized at $\sigma^{\star}\propto n_{\text{loc}}^{-1/(4+D')}$, yielding $\mathrm{MSE}=O(n_{\text{loc}}^{-4/(4+D')})$. Equivalently, to achieve an $\varepsilon$-accurate estimate one needs $n_{\text{loc}}=\Omega(\varepsilon^{-(4+D')/2})$, which scales only with the *interpretable* dimension $D' \ll D$. Additionally when $D'$ is not small relative to $n_{\text{loc}}$, we apply Ledoit–Wolf shrinkage to the local covariance. (Ledoit & Wolf, 2004) prove this remains consistent even if $D'/n_{\text{loc}}>0$, and crucially PatternLocal never needs an explicit matrix inverse, because Eq. 7 divides by the *scalar* variance of $\hat y$. Finally, recent locality-aware sampling theory shows that GLIME (Tan et al, 2023) needs only $n_{\text{loc}} = O(D'\log D')$ samples for a stable linear fit. PatternLocal can inherit the same sampling strategy, further reducing the data required per explanation without altering its closed-form update.
>
> ---
>
> > "All experiments are primarily on synthetic, or semi-synthetic image data. The generalization to complex real-world datasets remains unclear … What kind of explanations does PatternLocal produce when be applied to datasets like ImageNet or tabular data, where finding sufficient, well-aligned local neighbors is difficult?"
> >
>
> We acknowledge that our evaluation datasets (XAI-TRIS and the lesion MRI) are controlled or semi-synthetic. The choice was deliberate: these benchmarks provide *ground-truth feature importance* to quantitatively evaluate suppressor mitigation. As mentioned in **Limitations,** for real-life structured domains like medical imaging, where standardized views are common, we expect the PatternLocal method to perform well. In the setting of ImageNet, we expect the PatternLocal to have a harder time as there might be too few samples in the local neighborhood.
>
> We have added example with EEG signal on Event-Related Potential (ERP) tasks. More specifically EEG Motor Movement Dataset, where participants either move their left or right hand at some stimulus. Here we know that no pre-stimulus signal is correlated with the target variable, however we observe common XAI methods highlight pre-stimulus activity while PatternLocal does not. This is a real-life dataset where PatternLocal outperforms other methods.
>
> ---
>
> > "Why focus on Earth Mover's Distance (EMD) and Importance Mass Error (IME) and not simply using more standard metrics like MSE or PSNR for visual explanation comparison? How sensitive are your conclusions to metric selection?"
> >
>
> We chose EMD and IME because they both de-emphasize the impact of false-negative omissions of features in the ground truth on performance, while emphasizing the impact of false-positive attributions of importance to pixels not contained in the ground truth (Clark et al, 2024). We have added a motivation for these metrics in Section 4.4. In addition, we already do include the MSE metric in Appendix E: Ablation Studies, which still demonstrates the advantages of using PatternLocal. Thus we conclude that it is not sensitive to metric choice. As requested by mentioned by Reviewer Xj2D, we also include discussion on insertion/deletion. We have also added metrics on local fidelity of the surrogate model as the weighted error as requested by Reviewer zNrP. See answer to Reviewer Xj2D for a discussion of faithfulness metrics, which we believe is not a good metric for ground truth evaluation.
>
> ---
>
> > "The method appears to be computationally way more expensive compared to the baselines and the experimental results only partly justify the effort despite extensive hyperparameter search … The paper uses exhaustive hyperparameter tuning for all methods. Is this necessary for PatternLocal to perform well, or do optimal hyperparameters at least generalize in some way? How does PatternLocal perform in more typical, default or lightly-tuned scenarios compared to common default settings of existing explanations?"
> >
>
> We now provide runtime measurements (see table below). PatternLocal adds one extra weighted regression per explanation on top of the cost of the local surrogate. The computationally heavy part is fitting the initial surrogate model with methods like LIME and the extra regression keeps the runtime in the same order of magnitude. As PatternLocal only regresses in the local neighborhood, it is in practice also drastically improved by only considering samples with a weight above a certain threshold. Besides this, to mitigate this issue, like LIME, we already use superpixels to reduce the computational cost and, and have also included the LowRank approximation as seen in Appendix D: Extended results. Given that the runtime is only slightly higher than comparable model-agnostic methods like LIME, KernelSHAP, etc. but consistently out performs them, we believe that it is more than justified. Even though it is difficult to compare runtimes due to different hardware, implementations and hyperparameters. Here we have kept everything the same and averaged the runtime over 1000 instances of the XAI-TRIS dataset on an AMD EPYC 9124 16c/32t 3.0 GHz (also for model inference by LIME).
>
> | XAI method | Average runtime ± std (s) |
> | --- | --- |
> | PatternLocal | 12.9 ± 1.5 |
> | LIME | 7.4 ± 0.3 |
> | PatternLocal + Superpixel | 3.6 ± 0.3 |
> | LIME + Superpixel | 1.4 ± 0.1 |
> | PatternLocal + LowRank | 7.7 ± 0.4 |
> | LIME + LowRank | 6.9 ± 0.2 |
>
> We performed an exhaustive search for every method solely to avoid putting the baselines at a disadvantage; PatternLocal inherits the sensitivity to hyperparameters that LIME is known to have. If you look at the hyperparameter space Appendix C.2, PatternLocal only requires bandwidth, kernel and regularization parameter. For the XAI-TRIS dataset the kernel choice was nearly always the Gaussian kernel and regularization parameter was near to none, thus it was really only the bandwidth that made the difference. Choosing the bandwidth as the median pair-wise distance in the neighborhood also yields favorable results for PatternLocal. We therefore conclude that PatternLocal is not overly sensitive to hyperparameters; nevertheless, we include guidance in the appendix on choosing them.
>
> ---
>
> > "Could access to generative models or learned latent spaces provide a substitute for local neighborhood samples where data is limited or not aligned? Would such extensions preserve PatternLocal's suppressor-mitigation properties?"
> >
>
> This is an insightful suggestion, and we have added a paragraph discussing it. PatternLocal relies on a representative local neighborhood to estimate the forward pattern. When such neighborhoods are scarce, one could sample neighbors from a generative model conditioned on $\mathbf{x}_\star$, e.g., a VAE or a diffusion model, to augment the data. We note, however, that generative models may introduce bias if the latent space does not faithfully capture the local data distribution. Suppressor mitigation depends on accurate (explicit or implicit) estimates of the covariance between simplified features and the surrogate output; sampling from an unaligned latent space could reintroduce spurious correlations. We therefore consider integration with generative models as promising future work ,but beyond the scope of this paper.
>
> ---
>
> ## Final note
>
> We thank Reviewer zNrP once more for their constructive feedback. As detailed above, we have answered every question, clarified all ambiguities and updated the manuscript accordingly. We therefore kindly ask you to reconsider your scores in light of these improvements. Should any further points arise, we are happy to provide additional clarification at any time.

---

> ### Comment · Reviewer_zNrp · 2025-08-04
>
> I would like to thank the authors for their detailed response. Many of my concerns have been addressed, and I have accordingly increased my score. However, I still have some doubt regarding the overall utiliy of the method as requires standardized views and the existence of sufficiently similar samples in the local neighborhood.  Providing more evidence on how the method can be applied more broadly would substantially strengthen the paper.

---

> > ### Author Response · Authors · 2025-08-07
> >
> > We thank you for the response and for increasing your score. We agree that we need evidence beyond semi-synthetic images, which is why we added the experiment on EEG Motor-Movement dataset, a real-world, high-variance biomedical signal in which only the time axis is aligned across trials. In this setting, standard explainers (LIME, KernelSHAP, gradients) routinely highlight pre-stimulus activity that is known to be unrelated to hand movement, whereas PatternLocal leaves this baseline period dark and instead focuses on the expected post-stimulus sensorimotor rhythm. Because each subject provides only a few dozen trials, the neighborhoods are still rather sparse, confirming that PatternLocal is still reliable when local samples are limited.
> >
> > The EEG example also demonstrates another modality and highlights the importance of choosing a modality-appropriate definition of neighborhood. We have added the main results and a brief explanation to Section 4, with details in the appendix.

---

### Official Review · Reviewer_zrTT · 2025-07-02

**Clarity:** 1
**Significance:** 2
**Originality:** 3
**Rating:** 4
**Confidence:** 3

**Summary:**

The authors investigate the problem of XAI methods mis-attributing *suppressor variables*; that is, variables that affect model prediction but do not have statistical dependency with the target. An existing method, *Pattern*, can remove the effect of suppressor variables for feature attribution, however this method is global and therefore may not work well when applied to complex, nonlinear models. To solve this problem, the authors propose, *PatternLocal*, which extends  Pattern to be able to work on local neighborhoods in conjunction with locally discriminative surrogate models such as KernelSHAP or LIME. The proposed method is evaluated on the XAI-TRIS synthetic dataset and an MRI benchmark dataset.

**Questions:**

* The prediction model is effectively a surrogate of the data generating process. The attribution method is then a surrogate of the prediction model (which is also a surrogate). PatternLocal adds another layer on top of that. Have you evaluated how accurate PatternLocal is with respect to the model prediction or ground truth target? For example, many works on attribution methods use faithfulness or fidelity metrics to evaluate their performance [1].

[1] https://arxiv.org/pdf/2211.05667

**Ethical Concerns:**

["NO or VERY MINOR ethics concerns only"]

**Final Justification:**

My initial score was 3. The authors' responses addressed most of my concerns around motivation and proposed changes to fix notation and add sample complexity results, therefore I increased my score to 4. It is difficult to recommend a score of 5, since one of the original concerns was writing quality and we cannot see the revised manuscript.

**Limitations:**

Limitations are addressed on Page 9.

**Quality:**

2

**Strengths And Weaknesses:**

**Strengths:**
* The proposed method is a natural extension of Pattern, and is fairly intuitive.
* The experimental results on synthetic datasets show that PatternLocal outperforms competing methods in suppressing attributions for suppressor variables.

**Weaknesses:**
* Writing Quality and Notational Issues. I found the manuscript unnecessarily difficult to understand. In particular, the notation and writing could be more rigorous (see **Minor Issues** and **Questions** below). It would be helpful if there was a written algorithm that summarized PatternLocal. Figure 1 is very confusing -- it is not clear what the different colors, dotted lines, marker shapes, heatmaps, etc. represent, since they are not defined.
* Motivation. The purpose of most attribution methods (e.g. LIME, KernelSHAP) is to identify the most relevant features for the model prediction. If the model uses a suppressor variable in its prediction, then assigning that variable a nonzero attribution seems like expected behavior. It is not clear to me why a user of LIME or KernelSHAP would want to suppress attributions suppressor variables rather than training a better model. It would be helpful if the authors could add more discussion in the manuscript regarding motivation and use-cases.
* Missing related works section. It would be helpful to have more discussion around related works.
* Limited theoretical contribution. The paper could be improved if there was some theoretical results, e.g. sample complexity (especially since one of the stated limitations is having enough training data in the neighborhood of each explanation).

**Miscellaneous Minor Issues / Questions**
* Line 64: "Throughout we therefore restrict…"
* Line 125 says "regressing the local surrogate output… onto the neighborhood of x", but line 128 says "by regressing the simplified input x on the surrogate model's response…". These two lines seem to contradict each other.
* The variable $s$ seems to be overloaded -- it's sometimes used to represent a latent vector (equation 1), and sometimes it's a function (equation 6) which is also equal to a? In Eq. 10, $s$ is also an importance map.
* In equation 5,  $\mathcal{Z}$ is treated as a distribution over which you take an expectation, but it's defined as a "simplified input space". There is a similar issue in equation 6 for $\mathcal{X}$.
* Equation 6 has the term: $(h_{x}(x) - u \tilde y)^2$. Is u a D'-dimensional vector or a scalar? Should there be a $l_1$ or $l_2$ norm here?
* $\mathcal{F}^+$ in equation 10 is not defined. Is this supposed to be the same as $F^+$?
* In Section 4.4, what does $|s|$ represent? Equation 10 seems to imply that it's an absolute value or normalization, but Equation 11 implies that $|s|$ is an integer.

---

> ### Author Rebuttal · Authors · 2025-07-31
>
> # Reviewer zrTT
>
> Thank you for the detailed and constructive review. We greatly appreciate that you found the problem of suppressor variables important, the extension of the Pattern principle to a local setting novel and intuitive, and the empirical gains on suppressor benchmarks compelling. **Note, some subscripts were omitted to avoid OpenReview formatting issues.**
>
> ### Questions and Comments
>
> ---
>
> > "Writing Quality and Notational Issues. I found the manuscript unnecessarily difficult to understand … It would be helpful if there was a written algorithm that summarized PatternLocal."
> >
>
> We’ve updated Section 3: PatternLocal and added a high-level algorithmic description at the start (see response to Reviewer 7PSJ). This gives the reader a quick overview before the discussion of advantages like suppressor-variable mitigation, model faithfulness, local fidelity, and feasibility. We’ve also included a notation table in the appendix (see response to Reviewer 7PSJ), referenced in Section 2.1. These changes help clarify the manuscript and address the earlier notation issues.
>
> ---
>
> > "Figure 1 is very confusing ... "
> >
>
> We appreciate this feedback. In the revision the caption for Figure 1 now explicitly describes each element: colors denote class labels in the 2D projection, solid versus dotted lines indicate the model's decision boundary and the local linear surrogate, markers indicate sampled neighbors and the test point $\mathbf{x}_\star$, and the heatmaps show the resulting importance maps from the discriminative surrogate and the generative PatternLocal transformation. To avoid clutter we reduced the number of graphical elements and added a legend.
>
> ---
>
> > "Motivation. The purpose of most attribution methods (e.g. LIME, KernelSHAP) is to identify the most relevant features for the model prediction. If the model uses a suppressor variable in its prediction, then assigning that variable a nonzero attribution seems like expected behavior. It is not clear to me why a user of LIME or KernelSHAP would want to suppress attributions suppressor variables rather than training a better model. It would be helpful if the authors could add more discussion in the manuscript regarding motivation and use-cases."
> >
>
> We agree that standard XAI methods should show which features a model uses, but our goal is to spotlights those that are also statistically linked to the outcome. The difference lies between features the model exploits and features that truly matter for the target. Suppressor variables may raise accuracy by canceling noise yet hold no real link to the target; explanations should hide or flag them so users are not misled.
>
> Training a stronger model may not remove suppressors, especially in complex domains where the optimal model may always need to latch onto context information to remove variance from target-related variables. PatternLocal tackles this by turning the surrogate’s local model into a generative explanation aligned with the data distribution, thus down-weighting features unconnected to the target.
>
> We now underscore this point in the Introduction and Section 2.2 and outline domains, such as science and medicine, where suppressor insensitive explanations gives more actionable insights, whereas highlighting a suppressor could misleed. See the following added remark:
>
> Remark: It can be desirable for an optimal model to use suppressor variables in its prediction, however for the later explanations, we wish to filter-out these suppressor variables as they have no direct influence on the target variable in the data generating process nor any other statistical association (be it causal, anti-causal or through confounding). It is also not to be confused with the Clever Hans Effect or Shortcut learning where a model uses undesirable features.
>
> ---
>
> > "Missing related works section. It would be helpful to have more discussion around related works."
> >
>
> The related work was previously part of the Introduction, but we've now added a dedicated section. In addition to the original Pattern method, PatternNet, PatternAttribution, and local explainers like LIME and KernelSHAP, we now cover concept-based methods (e.g. TCAV), generative approaches like autoencoder-based explanations, and other suppressor-mitigation techniques. We also relate our work to causal attribution, connecting it to faithfulness metrics, infidelity, insertion/deletion scores (Chen et al., 2024).
>
> New source added: What makes a Good?: A harmonized view of properties of explanations by Chen et al., 2024.
>
> ---
>
> > "Limited theoretical contribution. The paper could be improved if there was some theoretical results, e.g. sample complexity ..."
> >
>
> While PatternLocal builds on the insight of transforming discriminative weights into generative patterns, extending this idea to local surrogates requires a non-trivial adaptation. The paper derives the kernel-weighted regression objective (Eq. 6) and its closed-form solution for $\ell_2$ regularization (Eq. 7). We further have added a discussion on sample complexity which can be seen under the response to Reviewer zNrP. These theoretical additions clarify how PatternLocal generalizes Pattern to non-linear settings and highlight its novelty.
>
> ---
>
> > "Line 125 says 'regressing the local surrogate output… onto the neighborhood of x', but line 128 says 'by regressing the simplified input x on the surrogate model's response' ..."
> >
>
> We appreciate the reviewers taking time to highlight this issue. Line 125 has been changed and should of course say *"PatternLocal extends this idea to the local setting by regressing the neighborhood of $x_\star$ onto the local surrogate output $\hat{y} = \mathbf{w}^\top \mathbf{x}$, thus removing suppressor effects specific to the instance."*
>
> ---
>
> > "In equation 5, $\mathcal{Z}$ is treated as a distribution over which you take an expectation, but it's defined as a "simplified input space". There is a similar issue in equation 6 for $\mathcal{X}$… Equation 6 has the term: $(h_{\mathbf{x}_\star}(\mathbf{x}) - \mathbf{u} \hat{y})$ . Is u a $D'$-dimensional vector or a scalar? Should there be a or norm here?"
> >
>
> Thank you for pointing out the notational ambiguity. In the original text, $\mathcal{Z}$ and $\mathcal{X}$ were used both as sets and random variables. To clarify, we now introduce explicit probability measures: $P_{\mathcal Z}$ is the sampling distribution over the simplified input space $\mathcal Z$ (uniform Bernoulli in basic LIME), and $P_{\mathcal X}$ is the data distribution on $\mathcal X \subseteq \mathbb{R}^{D}$, approximated by the training set.
>
> You're also right that it should be a norm. This was fixed right after submission and already appears correctly in Appendix A.
>
> We'll also add a clarifying note in Section 2.1. These updates match the notation in GLIME (Tan et al., 2023) and address the issue raised.
>
> ---
>
> > The variable $\mathbf{s}$ seems to be overloaded -- it's sometimes used to represent a latent vector (equation 1), and sometimes it's a function (equation 6) which is also equal to a? In Eq. 10, is also an importance map.
> >
>
> We understand some of the confusion. In Eq. 1, $\mathbf{s}$ is a latent vector, which is to be consistent with the notation of (Haufe et. al. 2014). To avoid confusion, we have changed the symbol of the latent vector in the entire Section 2.2 (Suppressor variables), Section 2.3 (Forward vs. backward models) and Section 3.2 (Toy example) to use $\mathbf{m}$ instead.
> In the rest of the manuscript, $\mathbf{s}$, is consistently used to refer to importance map / saliency maps for different XAI methods. We use superscript, to indicate which one we are referring to e.g. $\mathbf{s}^\text{Pattern}$, $\mathbf{s}^\text{LIME}$, $\mathbf{s}^\text{PatternLocal}$. We have changed the notation to $\mathbf{s}(\mathbf{x}\star)$ to remind of the dependency on $\mathbf{x}_\star$.
>
> ---
>
> > $\mathcal{F}^+$ in equation 10 is not defined. Is this supposed to be the same as ? … In Section 4.4, what does $|\mathbf{s}|$ represent? Equation 10 seems to imply that it's an absolute value or normalization, but Equation 11 implies that is an integer.
> >
>
> We apologize for the oversight. We have fixed the issue. See response to Reviewer Xj2D for details.
>
> > The prediction model is effectively a surrogate of the data generating process. The attribution method is then a surrogate of the prediction model (which is also a surrogate). PatternLocal adds another layer on top of that. Have you evaluated how accurate PatternLocal is with respect to the model prediction or ground truth target? For example, many works on attribution methods use faithfulness or fidelity metrics to evaluate their performance.
> >
>
> We partially agree with the first statement. We would rather say that PatternLocal updates the same surrogate model to de-emphasize suppressor variables. We believe this is a necessary approach and as mentioned in the response to Reviewer zNrP, this does not influence running time much.
>
> In regards to the metric, we have added metrics on local fidelity of the surrogate model as the weighted error as also requested by Reviewer zNrP. We do not believe that faithfulness is a suitable metric for ground truth evaluation and as it can lead to misleading results. See answer to Reviewer Xj2D regarding a discussion on faithfulness metrics.
>
> ## Final note
>
> We thank Reviewer zrTT once more for their constructive feedback. As detailed above, we have answered every question, clarified all ambiguities and updated the manuscript accordingly. We therefore kindly ask you to reconsider your scores in light of these improvements. Should any further points arise, we are happy to provide additional clarifications at any time.

---

> > ### Comment · Reviewer_zrTT · 2025-08-03
> >
> > I thank the reviewers for their response. I appreciate the clarification of the motivation, the additional sample complexity results and the notational fixes. Given the proposed changes, I will increase my score to 4.

---

> > > ### Author Response · Authors · 2025-08-07
> > >
> > > Thank you very much for taking the time to read our manuscript and for raising your score. We are glad that the additional motivation, the sample-complexity discussion, and the notational corrections and believe the manuscript is richer for it. The sample complexity analysis has been added to the appendix, while we have used the additional page to, among others, improve clarity, extend motivation, and add a dedicated related works section.

---

### Note · Authors · 2025-08-13

We thank the reviewers for their constructive feedback. PatternLocal converts locally linear surrogates into generative patterns that suppress false-positive attributions from suppressor variables, enabling better post-hoc instance-level explanations of non-linear models.

**Main changes during rebuttal/discussion:** Added a 4-step algorithm, a notation table, and revised Figure 1. Reorganized Sec. 3; added theory on sample complexity in the interpretable space; clarified Figure 4’s $\alpha$/SNR axis; justified EMD/IME and added MSE and local-fidelity metrics; reported runtimes; and added an EEG Motor-Movement example (Sec. 4, Sec. 5, App.).

**Reviewer concerns and fixes:**
- **Reviewer zrTT:** Fixed notation; defined $F^+$ and $|\cdot|$; clarified $h(\cdot)$ and losses near Eq. 5–6; added the high-level algorithm (Sec. 3); reported local fidelity (Sec. 4.4; App.).
- **Reviewer zNrp:** Strengthened theory (Eq. 6–7) and sample-complexity theory (App.); added real-life EEG data beyond synthetic images (Sec. 4, Sec. 5, App.); reported runtimes and bandwidth/shrinkage guidance (App.).
- **Reviewer Xj2D:** Conducted experiments with $l_1$-LIME (App.); clarified that sparsity alone does not remove suppressors; justified EMD/IME (Sec. 4.4); added MSE and local fidelity; fixed $|\cdot|$ vs. cardinality ambiguity (Sec. 4.4); and explained why insertion/deletion metrics can reward suppressors and are therefore not suitable in this context (Sec. 4.4; App.).
- **Reviewer 7PSJ:** Fixed notation and clarity issues; clarified $\alpha$/SNR trends in Fig. 4; and reordered Sec. 3.

**Summary:** On XAI-TRIS, PatternLocal shows strong gains on LIN, XOR, and RIGID (Fig. 2, Fig. 4). On artificial-lesion MRI, it concentrates importance on lesion regions, while edge filters concentrate on backgrounds (Fig. 5). On EEG, it suppresses irrelevant pre-stimulus activity and highlights expected post-stimulus activity (Sec. 4, App.). The method is model-agnostic and adds only one local regression atop the surrogate, keeping runtime in the same order as LIME. The limitations are that PatternLocal requires representative local neighborhoods and some cross-instance alignment. The performance may degrade with unstable surrogates or unaligned natural images.

**Final:** These changes and additions address the reviewers’ concerns, and we believe the paper meets the standards of novelty, significance, technical soundness, clarity, and reproducibility.

---

### Decision · Program_Chairs · 2025-09-17

**Decision:**

Accept (poster)

**Comment:**

The paper addresses false positive attributions caused by suppressor variables in explanations of nonlinear models. It introduces PatternLocal, which fits a local linear surrogate, then converts its discriminative weights into a generative pattern using neighborhood statistics, reducing suppressor influence while preserving local fidelity.

Reviewers highlight a clear problem framing, a principled extension of Pattern to local nonlinear settings, and compelling empirical gains across suppressor aware metrics. The method is model agnostic, inexpensive beyond the surrogate fit, and reports local fidelity and runtime.

Reviewers note that the initial manuscript had notational inconsistencies and limited theory. Evaluation leans on synthetic or semi synthetic images, with performance depending on well aligned local neighborhoods and stable surrogates. Sensitivity to surrogate hyperparameters remains. Broader demonstrations on diverse real world modalities and stronger generalization guarantees could further strengthen the work.

The paper tackles a consequential XAI failure mode, offers a simple and sound transformation that integrates with standard local explainers, and shows consistent improvements with reasonable cost. The added EEG study and theory address key doubts. After discussion, reviewers increased scores and signaled consensus for acceptance. An EEG example and sample complexity analysis were added. Reviewers acknowledged improvements and raised scores.